# Reversal of cell, circuit and seizure phenotypes in a mouse model of *DNM1* epileptic encephalopathy

Katherine Bonnycastle [1,2,3,5] ✉, Katharine L. Dobson [1,2,3],
Eva-Maria Blumrich[1,2,3], Akshada Gajbhiye [4], Elizabeth C. Davenport [1,2,3],
Marie Pronot [1,2,3], Moritz Steinruecke [1,2,3], Matthias Trost[4],
Alfredo Gonzalez-Sulser [1,2,3] & Michael A. Cousin [1,2,3] ✉

Dynamin-1 is a large GTPase with an obligatory role in synaptic vesicle endocytosis at mammalian nerve terminals. Heterozygous missense mutations in the dynamin-1 gene (*DNM1*) cause a novel form of epileptic encephalopathy, with pathogenic mutations clustering within regions required for its essential GTPase activity. We reveal the most prevalent pathogenic *DNM1* mutation, R237W, disrupts dynamin-1 enzyme activity and endocytosis when overexpressed in central neurons. To determine how this mutation impacted cell, circuit and behavioural function, we generated a mouse carrying the R237W mutation. Neurons from heterozygous mice display dysfunctional endocytosis, in addition to altered excitatory neurotransmission and seizure-like phenotypes. Importantly, these phenotypes are corrected at the cell, circuit and in vivo level by the drug, BMS-204352, which accelerates endocytosis. Here, we demonstrate a credible link between dysfunctional endocytosis and epileptic encephalopathy, and importantly reveal that synaptic vesicle recycling may be a viable therapeutic target for monogenic intractable epilepsies.

Heterozygous pathogenic missense mutations in the *DNM1* gene result in a specific form of developmental epileptic encephalopathy, characterised by severe to profound intellectual disability, hypotonia and epilepsy[1–3]. Epilepsy in these individuals typically starts with infantile spasms progressing to Lennox-Gastaut syndrome. The *DNM1* gene encodes the large GTPase dynamin-1, a mechanochemical enzyme that undergoes a conformational change on GTP hydrolysis, providing force for the final stages of vesicle fission[4,5]. It has a modular structure with an N-terminal GTPase domain, followed by domains essential for self-assembly (middle and GTPase effector domains, GED), membrane lipid binding (pleckstrin homology, PH) and protein interactions

(C-terminal proline-rich domain). All domains perform key roles in mediating dynamin-1 function[6–9], however its GTPase activity is essential for the SV fission reaction during endocytosis[4,5]. To date, all identified pathogenic mutations in the *DNM1* gene are localised to either the GTPase or middle domain, with one each in the PH and GED domains[10,11]. Importantly, almost all individuals with these mutations in *DNM1* have intractable epilepsy[1], making the identification of novel therapeutic interventions an urgent unmet challenge.

Considering the essential role for dynamin-1 in SV endocytosis and the clustering of mutations within the GTPase domain in individuals with epileptic encephalopathy, a logical prediction is that defects

[1]Centre for Discovery Brain Sciences, Hugh Robson Building, George Square, University of Edinburgh, EH8 9XD Edinburgh, Scotland, UK. [2]Simons Initiative for the Developing Brain, Hugh Robson Building, George Square, University of Edinburgh, EH8 9XD Edinburgh, Scotland, UK. [3]Muir Maxwell Epilepsy Centre, Hugh Robson Building, George Square, University of Edinburgh, EH8 9XD Edinburgh, Scotland, UK. [4]Newcastle University Biosciences Institute, Faculty of Medical Sciences, NE2 4HH Newcastle upon Tyne, UK. [5]Present address: Service de Génétique Médicale, Centre Hospitalier Universitaire (CHU) Sainte-Justine, Université de Montréal, Montreal, QC, Canada. ✉e-mail: Katherine.Bonnycastle@gmail.com; M.Cousin@ed.ac.uk

in SV endocytosis underpin this disorder. However, to date, no determination of the role of these predicted dominant-negative GTPase domain mutations has been performed at the neuron, circuit or behavioural level. To investigate this, we generated a mouse carrying the most prevalent mutation in the *DNM1* gene (R237W).

In this work, heterozygous $Dnm1^{+/R237W}$ mice display defective SV endocytosis, altered neurotransmission and seizure-like activity. Importantly, these phenotypes are all reversed by the small molecule BMS-204352, which we reveal accelerates SV endocytosis. We therefore reveal a preclinical model and potential therapy that will provide impetus to future small molecule screening studies and clinical trials to generate interventions for *DNM1* epileptic encephalopathy.

## Results

### R237W dynamin-1 displays reduced basal GTPase activity

Heterozygous mutations in the *DNM1* gene give rise to a specific form of epileptic encephalopathy[1], however little is known regarding how these mutations translate into this neurodevelopmental disorder. To determine this, we investigated the most prominent pathogenic *DNM1* mutation in human disease—R237W[1,2]. The R237 residue is critical for GTPase function, directly participating in GTP binding and stabilising the transition state of this region during GTP hydrolysis[12]. Because of this key role, we reasoned that substitution of a larger residue such as R237W would disrupt GTPase activity. To test this, we assayed the GTPase activity of full-length dynamin-1 in a heterologous expression system. We chose to use the splice variant dynamin-1aa, since this version is the predominant isoform in mammalian brain[3]. This version of dynamin-1 was fused to the fluorescent protein mCerulean (Dyn1-mCer) and then immunoprecipitated using nanobodies against the mCer moiety. As a positive control, we assessed the ability of the K44A dynamin-1 mutant (which is deficient in both GTP binding and hydrolysis[13]) to hydrolyse GTP. When these experiments were performed, a level of baseline GTPase activity was discovered in immunoprecipitates of the mCer protein. However, this was increased two-fold in Dyn1$_{WT}$-mCer immunoprecipitates (Fig. 1a, b). In contrast, the Dyn1$_{K44A}$-mCer mutant displayed a significantly reduced ability to hydrolyse GTP (Fig. 1a). Importantly, Dyn1$_{R237W}$-mCer displayed reduced GTPase activity at a similar level to Dyn1$_{K44A}$-mCer (Fig. 1b). Therefore, the R237W mutation disrupts the GTPase activity of dynamin-1.

### R237W dynamin-1 is dominant-negative for SV endocytosis

Dynamin-1 GTPase activity is essential for SV endocytosis[5]. To determine whether Dyn1$_{R237W}$ exerts a dominant-negative effect on this process, and in an attempt to mimic heterozygous individuals with *DNM1* mutations, Dyn1-mCer mutants were overexpressed in primary cultures of hippocampal neurons, 2–3 fold in excess of endogenous dynamin-1 (Supplementary Fig. 1a,b). The genetically-encoded reporter synaptophysin-pHluorin (sypHy) was used to monitor activity-dependent SV recycling. SypHy is the SV protein synaptophysin that has an exquisitely pH-sensitive GFP inserted into a lumenal domain[14]. The acidic interior of SVs results in the quenching of sypHy fluorescence in resting nerve terminals. During SV exocytosis, the reporter is exposed to the extracellular space and the subsequent unquenching provides a readout of SV fusion. SypHy remains fluorescent during endocytosis and is quenched on acidification of the newly formed SV. The speed of SV endocytosis is rate limiting when compared to SV acidification[14,15], meaning that the loss of sypHy fluorescence is indicative of the rate of SV endocytosis.

Neurons were challenged with a train of 300 action potentials (10 Hz) with the total SV recycling pool revealed by subsequent application of an alkaline solution (NH$_4$Cl). Neurons expressing the mCer empty vector displayed a typical sypHy response, with a rapid increase in fluorescence (reflecting SV exocytosis) followed by a slow decrease (SV endocytosis, Fig. 1c). When neurons overexpressing

Dyn1$_{WT}$-mCer were monitored, there was no difference in either SV endocytosis (measured as the amount of sypHy left to retrieve (Fig. 1c, d), or SV exocytosis (measured as the extent of the evoked sypHy peak as a proportion of the total SV pool, Fig. 1e). In neurons expressing Dyn1$_{K44A}$-mCer, SV endocytosis was significantly retarded, whereas SV exocytosis was unaffected (Fig. 1c–e). When neurons overexpressing Dyn1$_{R237W}$-mCer were assessed, SV endocytosis was greatly reduced compared to Dyn1$_{WT}$-mCer control, with no significant effect on SV exocytosis (Fig. 1f–h). In contrast, overexpression of a middle domain dynamin-1 mutant (A408T), from the *Fitful* mouse[16] which has normal GTPase activity (Supplementary Fig. 1c), had no dominant-negative impact on either SV endocytosis or exocytosis (Supplementary Fig. 1d–f). Therefore, the pathogenic *DNM1* GTPase mutant R237W, has a selective, dominant-negative effect on SV endocytosis.

### $Dnm1^{+/R237W}$ mice display defective SV endocytosis

The overexpression of Dyn1$_{R237W}$-mCer in $Dnm1^{+/+}$ neurons does not accurately recapitulate the in vivo situation, where this mutation would be expressed via its endogenous locus. Furthermore, it does not allow a direct investigation of how reduced SV endocytosis could culminate in epileptic encephalopathy. To address this, we generated a mouse line that expressed a $Dnm1^{R237W}$ allele, using CRISPR-Cas9 technology. Heterozygous $Dnm1^{+/R237W}$ mice were fertile and were born in Mendelian proportions. No gross alterations in brain architecture were observed using Nissl staining (Fig. 2a). Furthermore, the $Dnm1^{+/R237W}$ mouse is not a hypomorph, since quantitative Western blotting and mass spectrometry analysis revealed no change in dynamin-1 expression in either primary hippocampal neurons from $Dnm1^{+/R237W}$ mice, brain lysates from either 3 week- or 6 week-old $Dnm1^{+/R237W}$ mice (Supplementary Fig. 2a, b) or $Dnm1^{+/R237W}$ nerve terminals (Supplementary Data 1).

Western blotting for common SV recycling proteins and dynamin-1 interaction partners in primary hippocampal neurons from $Dnm1^{+/R237W}$ mice revealed no differences in their protein levels (Fig. 2b, c). This was also the case in brain lysates from either 3 week-old or 6 week-old $Dnm1^{+/R237W}$ mice (Supplementary Fig. 2c–g). To determine more global changes in presynaptic protein expression, quantitative mass spectrometry was performed on nerve terminals isolated from either $Dnm1^{+/+}$ or $Dnm1^{+/R237W}$ littermates (Fig. 2d). We established a list of 4237 quantified proteins associated to synapses, mitochondria and vesicular structures and transport (Supplementary Fig. 3). This revealed 151 proteins that were significantly increased in $Dnm1^{+/R237W}$ nerve terminals, with 39 significantly decreased (Fig. 2e, Supplementary Data 1). To identify cellular functions that may be either upregulated or perturbed, network analysis using the STRING web tool (v.11.5) was performed on the proteins that were significantly altered. Upregulated proteins in $Dnm1^{+/R237W}$ nerve terminals clustered around cell functions such as the proteasome core complex and the ErbB signalling pathway, whereas downregulated proteins were associated with G-protein coupled receptor signalling pathway and the dynactin complex (Fig. 2f). Therefore, while there are no gross changes in architecture or protein expression in $Dnm1^{+/R237W}$ mice, subtle variations are present at their nerve terminals that may reflect either disrupted cell signalling or potential compensatory mechanisms.

When hippocampal brain sections of $Dnm1^{+/R237W}$ mice were examined at the ultrastructural level, their nerve terminals repeatedly displayed misshapen SVs and endosomal-like compartments, in contrast to $Dnm1^{+/+}$ controls (Fig. 3a). When quantified, there was a 250% increase in the number of presynaptic endosomes in $Dnm1^{+/R237W}$ nerve terminals when compared to littermate controls (Fig. 3b). There was also a small (17%) but significant increase in the number of SVs (Fig. 3c). SVs in $Dnm1^{+/R237W}$ nerve terminals were larger than $Dnm1^{+/+}$ controls (Fig. 3d), with no significant change in endosome area (Supplementary Fig. 4a). This morphological phenotype suggested

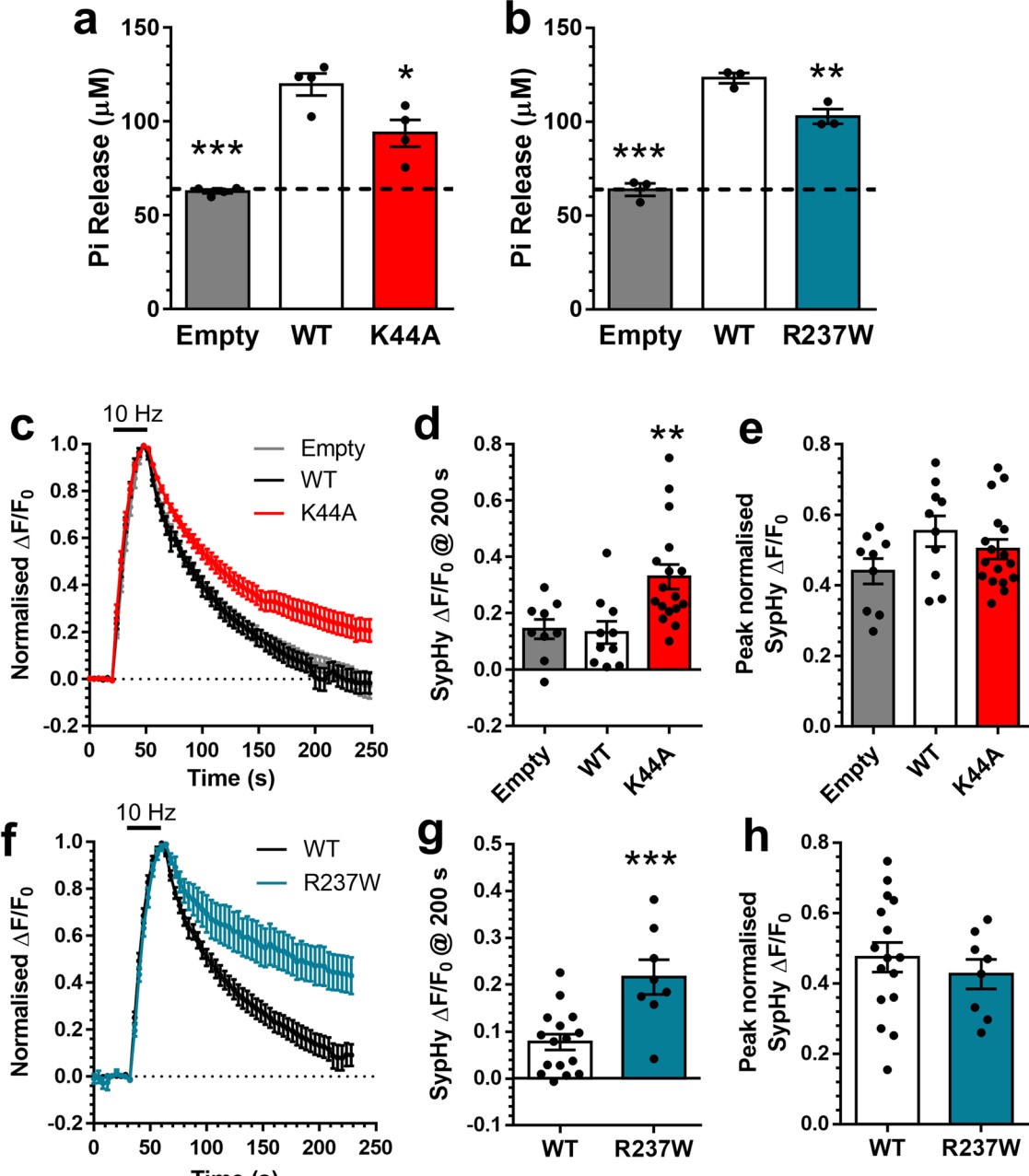

**Fig. 1 | The R237W *DNM1* GTPase domain mutation impairs SV endocytosis in a dominant-negative manner. a, b** HEK293T cells were transfected with either mCer (Empty), Dyn1$_{WT}$-mCer, Dyn1$_{K44A}$-mCer or Dyn1$_{R237W}$-mCer. After 48 h the cells were lysed and mCer was immunoprecipitated. The GTPase activity of the immunoprecipitate is displayed as released Pi ± SEM (one-way ANOVA, (**a**) all $n = 4$ separate experiments, ***$p < 0.0001$ WT to Empty, *$p = 0.0138$ WT to K44A; (**b**) all $n = 3$ separate experiments, ***$p < 0.0001$ WT to Empty, **$p = 0.0093$ WT to R237W). **c–h** Primary cultures of hippocampal neurons were transfected with synaptophysin-pHluorin (sypHy) and either mCer (Empty), Dyn1$_{WT}$-mCer, Dyn1$_{K44A}$-mCer or Dyn1$_{R237W}$-mCer between 11 and 13 DIV. At 13–15 DIV, cultures were stimulated with a train of 300 action potentials (10 Hz). Cultures were pulsed with NH$_4$Cl imaging buffer 180 s after stimulation. **c, f** Average sypHy response ($\Delta F/F_0 \pm$ SEM) normalised to the stimulation peak. Bar indicates stimulation (**c**, $n = 9$ Empty, $n = 10$ Dyn1$_{WT}$-mCer, $n = 17$ Dyn1$_{K44A}$-mCer; **f**, $n = 16$ Dyn1$_{WT}$-mCer, $n = 8$ Dyn1$_{R237W}$-mCer). **d, g** The average level of sypHy fluorescence ($\Delta F/F_0 \pm$ SEM) at 200 s (**d** one-way ANOVA, $n = 9$ Empty, $n = 10$ Dyn1$_{WT}$-mCer, $n = 17$ Dyn1$_{K44A}$-mCer, **$p = 0.0046$ WT to K44A; **g** Unpaired two-sided *t* test, $n = 16$ Dyn1$_{WT}$-mCer, $n = 8$ Dyn1$_{R237W}$-mCer, ***$p = 0.0007$). **e, h** The peak level of sypHy fluorescence ($\Delta F/F_0 \pm$ SEM) normalised to the NH$_4$Cl challenge (**e** one-way ANOVA, $n = 9$ Empty, $n = 10$ Dyn1$_{WT}$-mCer, $n = 17$ Dyn1$_{K44A}$-mCer, all ns; **h** Unpaired two-sided *t* test, $n = 16$ Dyn1$_{WT}$-mCer, $n = 8$ Dyn1$_{R237W}$-mCer, $p = 0.48$).

that there was a deficit in SV endocytosis, therefore primary hippocampal cultures from *Dnm1*$^{+/R237W}$ mice and *Dnm1*$^{+/+}$ littermates were prepared. As before, SV exocytosis and endocytosis were monitored using the sypHy reporter (Fig. 3e). Neurons from *Dnm1*$^{+/R237W}$ mice displayed a significant slowing in SV endocytosis during challenge with two different stimulus trains (300 action potentials at 10 Hz or 400 action potentials at 40 Hz, Fig. 3f, g, i, j). Furthermore, there

was no significant effect on SV exocytosis during either stimulus train (Fig. 3h, k). This was also the case when SV exocytosis was isolated in the presence of bafilomycin A1, which prevents acidification of SVs after endocytosis, removing contamination from retrieving SVs[17] (Supplementary Fig. 4b–e). Therefore *Dnm1*$^{+/R237W}$ neurons display a specific defect in SV endocytosis across a range of stimulus frequencies.

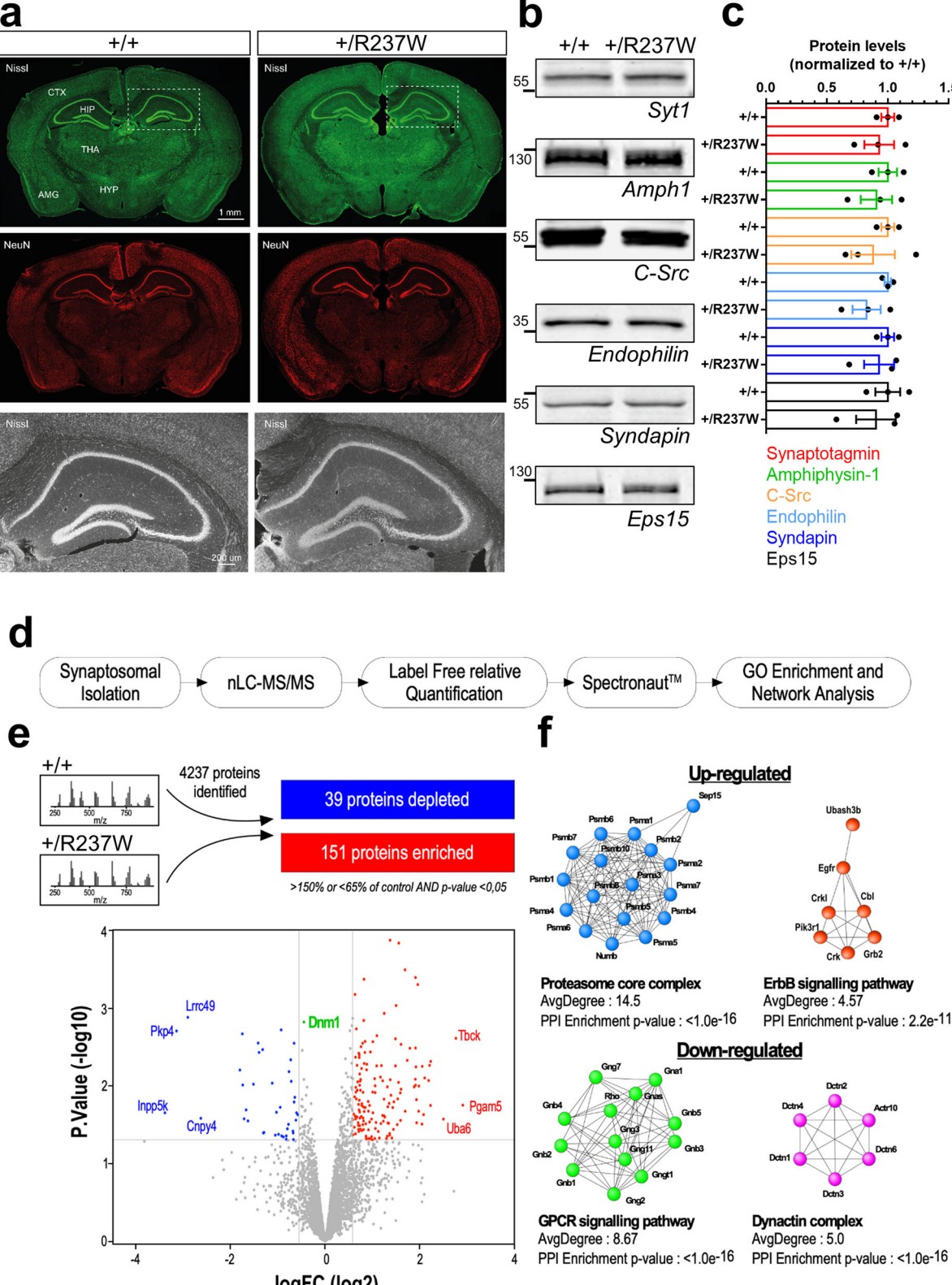

To confirm the endocytosis defect via a complementary approach, morphological analysis was performed using activity-dependent uptake of the fluid phase marker horseradish peroxidase (HRP). After its accumulation, HRP can be converted to an electron dense product, to reveal the number of endocytic intermediates generated during stimulation[18]. Recent studies have revealed that the majority of intermediates formed directly from the presynaptic plasma membrane are endosomes, which then shed SVs to refill the recycling pool[19,20]. $Dnm1^{+/R237W}$ neurons displayed a significant reduction in HRP-labelled endosomes compared to $Dnm1^{+/+}$ controls (Fig. 3l–n), with no change in HRP endosome size (Supplementary Fig. 4f), indicating activity-dependent generation of endosomes was impacted. Therefore $Dnm1^{+/R237W}$ neurons have an intrinsic and specific deficit in SV endocytosis.

**Fig. 2 | Dnm1⁺/R237W mice display no gross abnormalities but altered protein expression. a** Brains from 2 month-old *Dnm1⁺/⁺* and *Dnm1⁺/R237W* mice were perfusion-fixed and 50 μm brain sections were stained with Nissl and a NeuN antibody to label gross neuronal architecture. Cortex (CTX), thalamus (THA), hypothalamus (HYP), amygdala (AMG) and hippocampus (HPC) are labelled, scale bar 1 mm. Lower panels represent zoom images of the hippocampus, scale bar 200 μm. **b, c** Lysates from primary cultures of hippocampal neurons prepared from either *Dnm1⁺/⁺* and *Dnm1⁺/R237W* embryos were prepared and blotted for common presynaptic proteins and dynamin-1 interaction partners. **b** Representative blots display levels of Synaptotagmin-1 (Syt1), Amphiphysin-1 (Amph1), C-src, Endophilin, Syndapin and Eps15. **c** Quantification of protein levels normalised to *Dnm1⁺/⁺* ± SEM (*n* = 3 independent cultures for all, all ns, two-sided Mann-Whitney test). **d, e** Workflow of quantitative proteomic analysis. Total protein content was cleaned onto SDS-PAGE gel before tryptic digestion. Proteins were analysed by high-resolution tandem MS, with significant differences revealed using a two-sided unpaired *t* test corrected for multiple comparisons. **e** Volcano plot displays 4237 quantified proteins, 39 which were depleted in Dnm1⁺/R237W (blue), while 151 were enriched (red). Dnm1 level is displayed in green. **f** STRING network analysis of up- and down regulated proteins in Dnm1⁺/R237W synaptosomes. The resulting sub-complexes were subjected to MCL clustering at granularity 4, which results in 18 clusters including 2 clusters with an average superior to 4 for the up-regulated proteins (proteasome core complex, ErbB signalling pathways) and six clusters including two clusters with an average superior to 4 for the down-regulated proteins (GPCR signalling pathway, dynactin complex). Source data are provided as a Source Data file.

## Dnm1⁺/R237W mice have altered excitatory neurotransmission

SV endocytosis is essential to sustain neurotransmitter release[21,22], therefore we predicted that the endocytosis defects observed in *Dnm1⁺/R237W* neurons would translate into dysfunctional neurotransmission. To determine this, we examined neurotransmission at the excitatory Schaffer collateral CA3-CA1 synapse, using whole-cell patch clamp recordings. We first determined the intrinsic properties of *Dnm1⁺/R237W* neurons (Supplementary Table 1). Most parameters were unaffected, however differences in both Tau (membrane decay time) and capacitance (cell size) were detected. Alterations in capacitance may reflect dysfunctional endocytosis, whereas the elevated Tau value suggests the plasma membrane is slower to charge, reflecting a decrease in cell excitability. This increase in Tau is consistent with the increased half-width of action potentials in *Dnm1⁺/R237W* neurons, in addition to the decay rate and rise time (Supplementary Table 1). Therefore, *Dnm1⁺/R237W* neurons display alterations in their intrinsic properties, which may be an adaptation to defects at the cell or circuit level. However, there was no significant difference in the amount of current required to trigger action potential firing between *Dnm1⁺/R237W* and *Dnm1⁺/⁺* synapses when either spike frequency or rheobase was quantified (Fig. 4a–c, Supplementary Table 1). Therefore, the R237W *Dnm1* allele does not result in intrinsic hyperexcitability of *Dnm1⁺/R237W* neurons.

Next we investigated both miniature excitatory or inhibitory postsynaptic currents (mEPSCs, mIPSCs) at CA1 neurons, since these can reflect deficiencies in SV recycling. The frequency, but not amplitude, of mEPSCs was significantly reduced at *Dnm1⁺/R237W* synapses (Fig. 4d–f), suggesting a presynaptic SV recycling defect with no obvious postsynaptic phenotype. In contrast, we observed no significant difference in the frequency of mIPSC events and an increase in mIPSC amplitude at *Dnm1⁺/R237W* synapses (Fig. 4g–i). This suggests that there was a selective perturbation of excitatory neurotransmission in *Dnm1⁺/R237W* mice. To determine this, we assessed whether evoked excitatory or inhibitory neurotransmission was impacted in *Dnm1⁺/R237W* mice. Somewhat surprisingly, *Dnm1⁺/R237W* synapses displayed an increase in evoked EPSC amplitudes across a range of stimulus intensities (Fig. 4j, k). In contrast, *Dnm1⁺/R237W* synapses displayed no alteration in evoked IPSC amplitude across an identical stimulus range (Fig. 4l, m). Finally, we determined the paired-pulse ratio (PPR) for excitatory and inhibitory neurotransmission by applying pairs of pulses at a range of inter-stimulus intervals. This analysis revealed a significant decrease in the EPSC PPR and a significant increase the IPSC PPR in *Dnm1⁺/R237W* synapses when compared to *Dnm1⁺/⁺* (Fig. 5a–d). This suggests that the presence of the R237W *Dnm1* allele results in an increased release probability (Pr) for excitatory neurotransmission and decreased Pr for inhibitory neurotransmission, providing a potential mechanism for imbalanced excitability at these synapses.

Since *Dnm1⁺/R237W* synapses appear to have increased excitatory neurotransmission across a range of stimuli, we next investigated whether neurotransmission could be sustained during a prolonged train of high frequency action potentials (600 APs at 40 Hz). *Dnm1⁺/R237W* synapses displayed an inability to support neurotransmission during the stimulus train, when compared to *Dnm1⁺/⁺* controls (Fig. 5e–g). This finding, when considered with the reduced PPR, could be due to either increased Pr, or an inability to replenish SV pools. To determine this, the amplitude of the first evoked EPSC was divided by the effective readily releasable pool (RRP) size, estimated from the 40 Hz action potential train[23] (Fig. 5g). Pr was unaffected in *Dnm1⁺/R237W* circuits (Supplementary Fig. 5a), whereas both the size and replenishment rate of the RRP was significantly reduced (Fig. 5h,i). Therefore, excitatory neurotransmission in *Dnm1⁺/R237W* circuits is initially augmented (most likely via increased Pr), however this enhancement cannot be sustained during an action potential train (most likely via reduced SV endocytosis), resulting in its depression.

## Dnm1⁺/R237W mice display myoclonic jumping

Heterozygous mutations in the *DNM1* gene cause epileptic encephalopathies[1,2]. However many preclinical rodent models of monogenic epilepsies and neurodevelopmental disorders do not recapitulate the seizure activity observed in individuals with these disorders[24]. When examined, *Dnm1⁺/R237W* mice do not display overt spontaneous tonic-clonic seizures, however they do display "myoclonic jumping", which involves bursts of highly active jumping (Fig. 6a, Supplementary Movie 1). Importantly, in vivo electrophysiological recordings from *Dnm1⁺/R237W* mice during these myoclonic jumps revealed increased generalised spiking activity during these events (Fig. 6b). *Dnm1⁺/R237W* mice had increased power during "myoclonic jumping" events in the low-frequency delta and theta bands compared to the rare jumping events observed in *Dnm1⁺/⁺* mice (Fig. 6c, d). Low frequency bands are not associated with muscular activity, suggesting the increased power observed during myoclonic jumping is not a consequence of jumping per se. In addition, the electrophysiological activity associated with myoclonic jumping in *Dnm1⁺/R237W* mice was also longer than in *Dnm1⁺/⁺* mice (Fig. 6e). Taken together, this suggests that this behaviour is a consequence of hyperexcitability and potential seizure activity. Therefore, the *Dnm1⁺/R237W* mouse has both construct and face validity as a preclinical model of *DNM1* epileptic encephalopathy.

## BMS-204352 corrects phenotypes in Dnm1⁺/R237W mice

The presynaptic and circuit phenotypes observed in *Dnm1⁺/R237W* neurons are strongly supportive of dysfunctional SV endocytosis being the key driver of the myoclonic jumping observed in *Dnm1⁺/R237W* mice. Therefore, we next determined whether correction of SV endocytosis could restore normal neurotransmission and ablate the observed behavioural phenotypes. The small molecule BMS-204352 ((3S)-(+)-(5-chloro-2-methoxyphenyl)−1,3-dihydro-3-fluoro-6-(trifluoromethyl)−2H-indol-2-one) was chosen for this task since it is a therapeutic safe for use in humans[25] and can correct behavioural defects in a preclinical model of fragile X syndrome[26]. This latter effect prompted us to investigate its action, since a number of fragile X syndrome model systems display circuit hyperexcitability[27]. We first examined the effect of BMS-204352

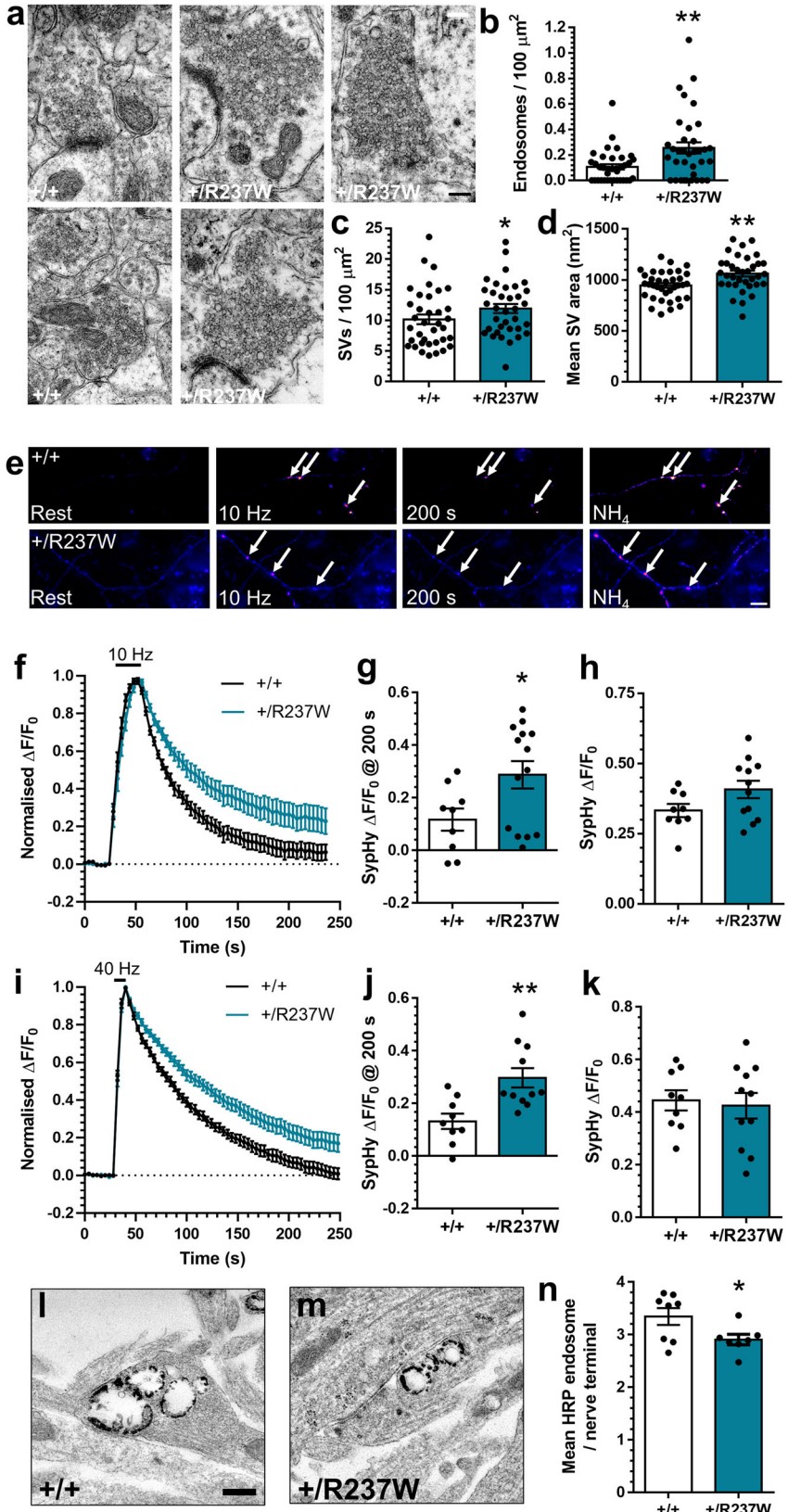

in primary cultures of *Dnm1*[+/+] hippocampal neurons overexpressing both Dyn1$_{WT}$-mCer and sypHy. Intriguingly, a dose-dependent acceleration of SV endocytosis was observed at time points after stimulation, with a reduction in SV exocytosis also observed at the highest dose (Supplementary Fig. 5b–d). Therefore BMS-204352 may have the potential to correct presynaptic defects in *Dnm1*[+/R237W] neurons.

BMS-204352 displays positive modulatory effects on both neuronal K$_v$7 channels and BK channels, whereas it is a negative modulator of both K$_v$7.1 channels and GABA$_A$ receptors[28,29]. This spectrum of activity across multiple potassium channel subtypes suggest that it may accelerate SV endocytosis via a series of different mechanisms. To determine this, we examined SV endocytosis and exocytosis in *Dnm1*[+/+]

**Fig. 3 | *Dnm1*<sup>+/R237W</sup> neurons display dysfunctional SV endocytosis. a** Brains from 2 month-old *Dnm1*<sup>+/+</sup> and *Dnm1*<sup>+/R237W</sup> mice were perfusion fixed and processed for electron microscopy. Representative images reveal enlarged endosomes in *Dnm1*<sup>+/R237W</sup> excitatory hippocampal nerve terminals, scale bar 250 nm. The number of presynaptic SVs (**b**) and endosomes (**c**) and size (**d**) of SVs were quantified ± SEM (*n* = 36 profiles *Dnm1*<sup>+/+</sup>, *n* = 35 *Dnm1*<sup>+/R237W</sup>, **b** *p* = 0.005, **c** *p* = 0.044 Mann-Whitney test, **d** *p* = 0.0024 Unpaired two-sided *t* test). **c–i** Primary cultures of hippocampal neurons prepared from either *Dnm1*<sup>+/+</sup> and *Dnm1*<sup>+/R237W</sup> embryos were transfected with synaptophysin-pHluorin (sypHy) between 7 and 9 DIV. At 13–15 DIV, cultures were stimulated with a train of either (**f–h**) 300 action potentials (10 Hz) or (**i–k** 400 action potentials (40 Hz). Cultures were pulsed with NH₄Cl imaging buffer 180 s after stimulation. **e** Representative images of the sypHy response in *Dnm1*<sup>+/+</sup> and *Dnm1*<sup>+/R237W</sup> neurons are displayed at Rest, during 10 Hz stimulation, at 200 s and during NH₄Cl. Arrows indicate responsive nerve terminals. Scale bar 10 μm.

**f, i** Average sypHy response (Δ*F*/*F*₀ ± SEM) normalised to the stimulation peak (**f**, *n* = 9 *Dnm1*<sup>+/+</sup>, *n* = 12 *Dnm1*<sup>+/R237W</sup>; **i**, *n* = 9 *Dnm1*<sup>+/+</sup>, *n* = 11 *Dnm1*<sup>+/R237W</sup>). **g, j** Average level of sypHy fluorescence (Δ*F*/*F*₀ ± SEM) at 200 s (**g** Two-sided Mann-Whitney test, *n* = 9 *Dnm1*<sup>+/+</sup>, *n* = 12 *Dnm1*<sup>+/R237W</sup> *\*p* = 0.045; **j** Unpaired two-sided *t* test, *n* = 9 *Dnm1*<sup>+/+</sup>, *n* = 11 *Dnm1*<sup>+/R237W</sup> **\*\*p* = 0.003). (**h, k**) Peak level of sypHy fluorescence (Δ*F*/*F*₀ ± SEM) normalised to the NH₄Cl challenge (**h** Unpaired two-sided t test, *n* = 9 *Dnm1*<sup>+/+</sup>, *n* = 12 *Dnm1*<sup>+/R237W</sup> *p* = 0.237; **k** Unpaired two-sided t test, *n* = 9 *Dnm1*<sup>+/+</sup>, *n* = 11 *Dnm1*<sup>+/R237W</sup> *p* = 0.751). **l–n** *Dnm1*<sup>+/+</sup> and *Dnm1*<sup>+/R237W</sup> neurons were stimulated with a train of 400 action potentials (40 Hz) in the presence of 10 mg/ml HRP. Representative images display HRP-labelled endosomes in *Dnm1*<sup>+/+</sup> (**l**) and *Dnm1*<sup>+/R237W</sup> (**m**) nerve terminals, scale bar 250 nm. **n** Average number of HRP-labelled endosomes per nerve terminal ± SEM (Unpaired two-sided *t* test, *n* = 8 *Dnm1*<sup>+/+</sup>, *n* = 7 *Dnm1*<sup>+/R237W</sup> *\*p* = 0.039). Source data are provided as a Source Data file.

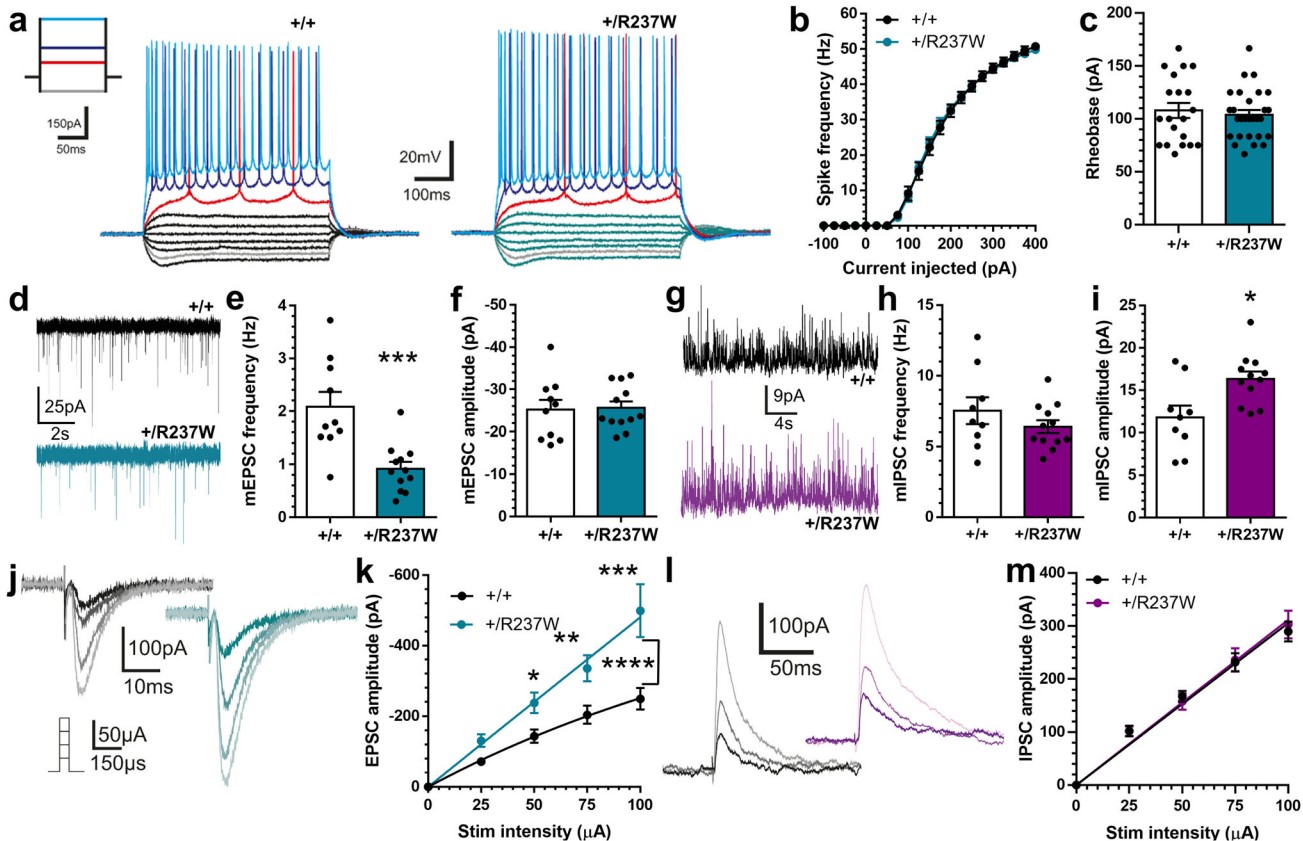

**Fig. 4 | *Dnm1*<sup>+/R237W</sup> mice display dysfunctional excitatory neurotransmission.** Neurotransmission at CA3/CA1 synapses was monitored using whole-cell patch clamp recording in acute hippocampal slices from *Dnm1*<sup>+/+</sup> and *Dnm1*<sup>+/R237W</sup> mice. **a** Representative voltage responses of CA1 pyramidal neurons in response to 500 ms hyper/depolarising current injections, with average spike frequency (**b**) and rheobase (**c**) ±SEM (*n* = 20 *Dnm1*<sup>+/+</sup>, *n* = 28 *Dnm1*<sup>+/R237W</sup>, **b** Two-way ANOVA, *p* = 0.997, **c** Unpaired two-sided *t* test, *p* = 0.613). **d** Example mEPSC events. Average frequency (**e**) and amplitude (**f**) of mEPSC events ± SEM (Two-sided Mann-Whitney test, *n* = 10 *Dnm1*<sup>+/+</sup>, *n* = 12 *Dnm1*<sup>+/R237W</sup>, **b** *p* = 0.0008, **c** *p* = 0.665). **g** Example mIPSC events. Average frequency (**h**) and amplitude (**i**) of mIPSC events ± SEM (Two-sided

Mann-Whitney test, *n* = 9 *Dnm1*<sup>+/+</sup>, *n* = 12 *Dnm1*<sup>+/R237W</sup>, **h** *p* = 0.379, **i** *p* = 0.023). **j–m** Acute hippocampal slices were stimulated at a range of intensities (25, 50, 75, and 100 μA, three repeats at each intensity, frequency 0.05 Hz) in a pseudo random order. Representative traces (**j**) and evoked EPSC amplitude ± SEM (**k**) is displayed (Two-way ANOVA with Fishers LSD, *n* = 34 *Dnm1*<sup>+/+</sup>, *n* = 39 *Dnm1*<sup>+/R237W</sup>, ****\*\*\*\*p* = 0.018, *\*p* = 0.043 50 μA, **\*\*p* = 0.005 75 μA, ***\*\*\*p* = 0.0004 100 μA). Representative traces (**l**) and evoked IPSC amplitude ± SEM (**m**) are displayed (Two-way ANOVA with Fishers LSD, *n* = 36 *Dnm1*<sup>+/+</sup>, *n* = 40 *Dnm1*<sup>+/R237W</sup>, overall and all pairwise comparisons *p* > 0.92).

hippocampal neurons expressing sypHy in the presence of a series of potassium channel modulators. These were: two structurally-unrelated BK channel agonists (NS11021 and BMS-191011), a BK channel antagonist (Paxilline), a K<sub>v</sub>7 channel activator (Retigabine), and a K<sub>v</sub>7 channel inhibitor (XE-991). No modulator was able to accelerate SV endocytosis in the manner observed with BMS-204352 (Supplementary Fig. 6). Therefore, the action of BMS-204352 on SV endocytosis is not due to modulation of a specific class of ion channels, suggesting its

presynaptic effects are an amalgamation of the modulation of some or all of these channels, or an as yet unidentified off-target effect.

Regardless of the BMS-204352 mechanism of action, we next examined whether it was able to correct defective SV endocytosis due to expression of the R237W dynamin-1 mutant. We first determined its effect on *Dnm1*<sup>+/+</sup> cells overexpressing Dyn1<sub>R237W</sub>-mCer. In these neurons, BMS-204352 fully restored SV endocytosis kinetics (Fig. 7a, b), suggesting it may be a viable intervention to correct dysfunction in

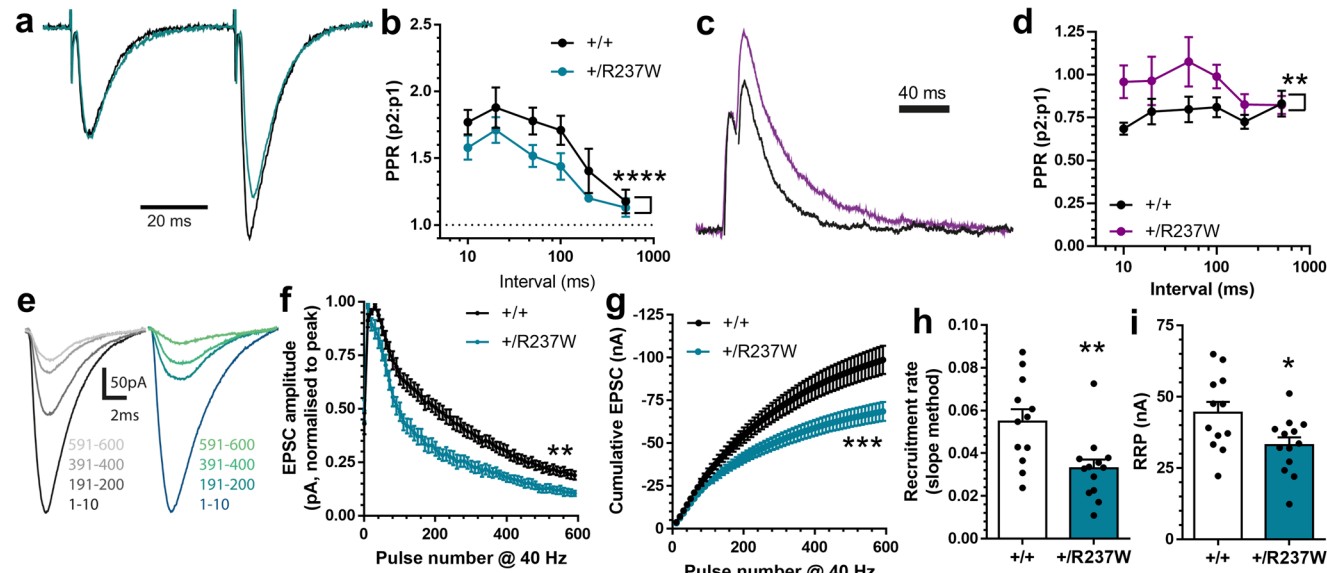

**Fig. 5 | *Dnm1*[+/R237W] mice display altered short-term plasticity.** Neurotransmission at CA3/CA1 synapses was monitored using whole-cell patch clamp recording in acute hippocampal slices from *Dnm1*[+/+] and *Dnm1*[+/R237W] mice. **a, b** Paired pulse ratio (PPR) of evoked EPSCs as a function of the inter-stimulus interval (10–500 ms) ± SEM (Two-way ANOVA with Fishers LSD, n = 11 *Dnm1*[+/+], n = 10 *Dnm1*[+/R237W], ****p < 0.0001). **c, d** PPR of evoked IPSCs as a function of the inter-stimulus interval (10–500 ms) ± SEM (Two-way ANOVA with Fishers LSD, n = 9 *Dnm1*[+/+], n = 13 *Dnm1*[+/R237W], **p = 0.0018). Slices were stimulated with 600 APs

(40 Hz). Representative traces (**e**) are shown as averages of ten consecutive responses from the pulse ranges stated. Average evoked EPSC amplitude (**f**) normalised to peak response is displayed ±SEM (Two-way ANOVA, n = 12 *Dnm1*[+/+], n = 13 *Dnm1*[+/R237W], **p < 0.0043). **g** Linear regression on the last 1 s of the cumulative EPSC plot in (**f**) ±SEM (Two-way ANOVA, ****p < 0.0001). The average rate of readily releasable pool (RRP) replenishment (**h**) and mean RRP size (**i**) ±SEM were estimated from the linear regression plot in (**g**). **h** Two-sided Mann-Whitney test, **p = 0.0045; **i** Unpaired two-sided *t* test, *p = 0.0229. Source data are provided as a Source Data file.

*Dnm1*[+/R237W] neurons. When the effect of BMS-204352 on SV endocytosis was examined in primary cultures of *Dnm1*[+/R237W] neurons, a full correction of SV endocytosis kinetics was again observed when compared to *Dnm1*[+/+] neurons (Fig. 7d, e). BMS-204352 had no significant effect on SV exocytosis in either *Dnm1*[+/+] neurons with overexpressed Dyn1[R237W]-mCer, or *Dnm1*[+/R237W] neurons (Fig. 7c, f). Therefore BMS-204352 restores SV endocytosis that was previously rendered dysfunctional via the mutant R237W *Dnm1* allele.

We next determined whether BMS-204352 could correct the observed dysfunction of excitatory neurotransmission in *Dnm1*[+/R237W] mice, since we predicted that defects in circuit activity were a result of impaired SV endocytosis. When applied to *Dnm1*[+/+] hippocampal slices, BMS-204352 had no effect on evoked EPSC amplitudes across a range of stimuli (Fig. 8a, b). However, BMS-204352 fully restored normal evoked EPSC amplitudes in *Dnm1*[+/R237W] slices to *Dnm1*[+/+] levels, across the same range of stimulus intensities (Fig. 8c, d). Therefore BMS-204352 can restore normal evoked excitatory neurotransmission in *Dnm1*[+/R237W] circuits.

We next examined whether BMS-204352 could reverse short-term plastic changes in excitatory neurotransmission by monitoring synaptic facilitation evoked via a 10 Hz AP train (15 s). In *Dnm1*[+/+] slices, a pronounced facilitation was observed (Fig. 8e, f), in agreement with previous studies[30]. In contrast, no facilitation of excitatory neurotransmission was observed in *Dnm1*[+/R237W] slices (Fig. 8e, f). Application of BMS-204352 to *Dnm1*[+/R237W] hippocampal slices fully restored facilitation to *Dnm1*[+/+] levels and had no effect on the *Dnm1*[+/+] response (Fig. 8e, f). Therefore BMS-204352 corrects fundamental defects in evoked excitatory neurotransmission and short-term plasticity in *Dnm1*[+/R237W] circuits.

Finally, we determined whether BMS-204352 was able to correct the myoclonic jumping phenotype in *Dnm1*[+/R237W] mice. *Dnm1*[+/R237W] mice and *Dnm1*[+/+] littermate controls were habituated in an open field arena for 30 min on day 1. This protocol was repeated for 5 days. On days 2 and 4, mice were dosed with either BMS-204352 or a vehicle control in a counterbalanced manner (Fig. 9a). Mice were also

monitored on days 3 and 5 to examine baseline behaviour (Washout, Fig. 9a). When baseline behaviour of *Dnm1*[+/+] and *Dnm1*[+/R237W] mice were examined, robust differences in both the number of myoclonic jumps and bursts of jumps (defined as a train of at least two myoclonic jumps with less than 2 s between consecutive jumps) were observed (Supplementary Fig. 7a, b). This phenotype was not due to increased general activity, since there is no significant change in the distance travelled by *Dnm1*[+/R237W] mice when compared to *Dnm1*[+/+] controls (Supplementary Fig. 7c).

During the test phase, the phenotypes that were observed during washout were retained in vehicle-treated *Dnm1*[+/+] and *Dnm1*[+/R237W] mice. Specifically, vehicle-treated *Dnm1*[+/R237W] mice displayed a significant increase in the total number of myoclonic jumps (Fig. 9b) and the number of jumping bursts (Fig. 9c) when compared to *Dnm1*[+/+] controls. Delivery of BMS-204352 to *Dnm1*[+/+] mice had no significant effect on these parameters (Fig. 9b, c). In contrast, BMS-204352 fully corrected both jumping phenotypes in *Dnm1*[+/R237W] mice to the levels observed in *Dnm1*[+/+] mice (Fig. 9b, c). Importantly, this correction was not due to depression of locomotive activity, since the distance travelled was not significantly different when *Dnm1*[+/R237W] mice treated with or without BMS-204352 were compared (Supplementary Fig. 7c). Furthermore, BMS-204352 had no effect on the time spent in the middle of the open area, indicating that the correction of seizure phenotypes was not due to previously documented anxiolytic effects of the molecule[29] (Supplementary Fig. 7d). In summary, these results suggest that BMS-204352 has high potential for therapy in *DNM1* epileptic encephalopathy, since it corrects dysfunction at the cellular, circuit and behavioural level in a preclinical model of this disorder.

## Discussion

Heterozygous *DNM1* mutations are responsible for a novel form of epileptic encephalopathy[1,2]. Here, we confirmed that the most common pathogenic *DNM1* mutation, R237W, disrupts dynamin-1 enzyme activity and SV endocytosis. Furthermore, using the *Dnm1*[+/R237W] mouse, we revealed that dysfunctional SV endocytosis translates into

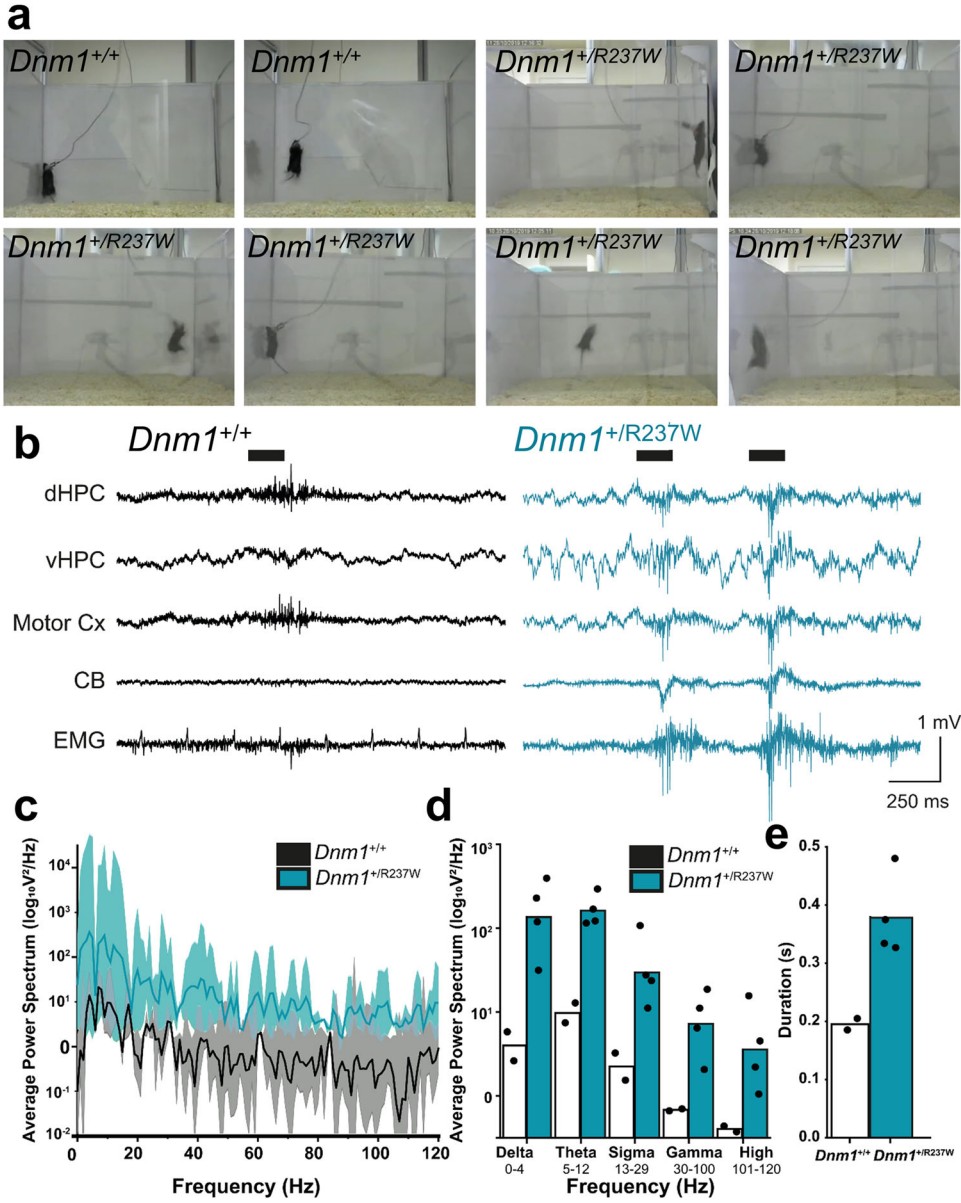

**Fig. 6 | *Dnm1*+/R237W mice display myoclonic jumping seizure-like activity.**
**a** Representative still images displaying the typical jumping behaviour of both *Dnm1*+/+ and *Dnm1*+/R237W mice. *Dnm1*+/+ mice occasionally jumped, however *Dnm1*+/R237W mice displayed stereotypical and burst-like events. **b** Example traces of in vivo LFP and electromyogram (EMG) recordings from dorsal hippocampus (dHPC), ventral hippocampus (vHPC), motor cortex (Motor Cx) and cerebellum (CB) during jumping activity in *Dnm1*+/+ and *Dnm1*+/R237W mice. **c** Power spectrum estimate across all jump epochs. Lines indicate mean values ± SEM. **d** Plot of average power in commonly used frequency bands during jumps. Bars indicate mean from $n = 2$ *Dnm1*+/+ and $n = 4$ *Dnm1*+/R237W. **e** Plot of average duration of electrographical activity associated with jumps. Bars indicate mean from $n = 2$ *Dnm1*+/+ and $n = 4$ *Dnm1*+/R237W. Source data are provided as a Source Data file.

altered excitatory neurotransmission and ultimately seizure-like phenotypes. Importantly, these phenotypes were corrected at the cell, circuit and in vivo level via the acceleration of SV endocytosis using BMS-204352. This study therefore provides a compelling link between dysfunctional SV endocytosis and epileptic encephalopathy, but moreover reveals that SV endocytosis may be a viable therapeutic route for monogenic intractable epilepsies.

The R237W mutation was chosen for our mouse model, since it is the most prevalent missense mutation in the *DNM1* gene (8 from 33 cases[1,2]). The *Dnm1*+/R237W mouse appears to have both face and construct validity and therefore is predicted to be of high value for future therapeutic studies. These mice displayed a selective defect in SV endocytosis, excitatory neurotransmission and a characteristic jumping phenotype. This behavioural phenotype occurred co-incident with increased generalised spiking activity, providing evidence that it may

be precipitated via seizure-like events. We named this jumping phenotype "myoclonic jumping" since it appears similar to phenotypes in several preclinical epilepsy models observed either in isolation or in progression towards full tonic-clonic seizures[31–33]. Furthermore, it is also observed in autism / neurodegeneration models as a measure of repetitive and stereotypic behaviour[34,35].

It is informative to contrast the *Dnm1*+/R237W mouse with a previously characterised mouse model of *DNM1* epileptic encephalopathy, the *Fitful* mouse[16]. This mouse does not model a human mutation, but instead arose from a spontaneous mutation (A408T) in the middle domain of the ax isoform of *Dnm1*, with mice homozygous for this mutation displaying spontaneous convulsive seizures resulting in lethality after 2–3 weeks[16]. Heterozygous *Fitful* mice also display spontaneous seizures that are detectable via EEG, or convulsive episodes on routine handling after 2–3 months. The A408T mutation

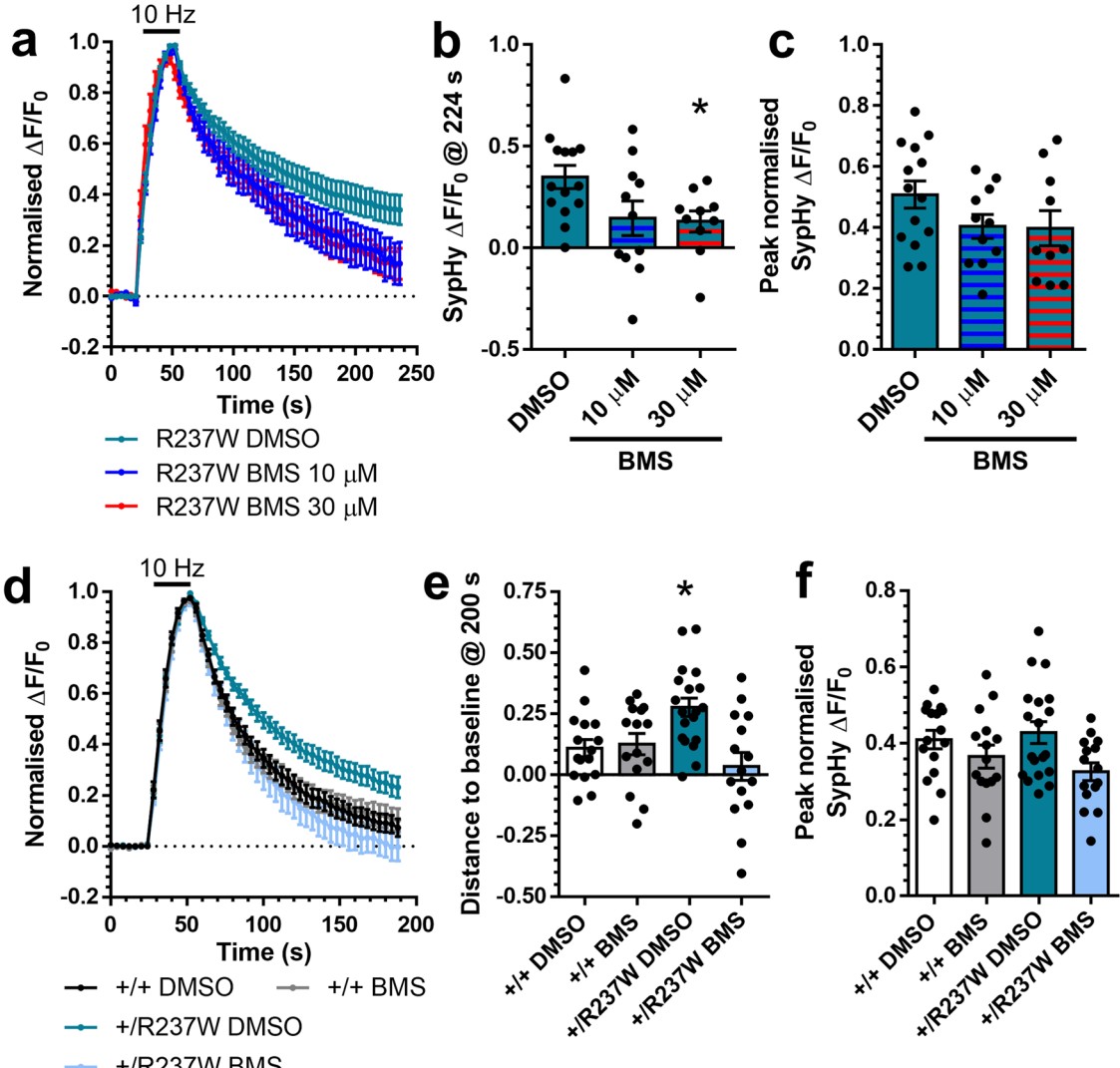

**Fig. 7 | BMS-204352 corrects dominant-negative effect of R237W mutation on SV endocytosis. a–c** Primary cultures of hippocampal neurons prepared from *Dnm1*[+/+] embryos were transfected with synaptophysin-pHluorin (sypHy) and Dyn1$_{R237W}$-mCer between 11 and 13 DIV. At 13–15 DIV, cultures were stimulated with a train of 300 action potentials (10 Hz) in the presence of either 10 µM or 30 µM BMS-204352 or a vehicle control (DMSO). Cultures were pulsed with NH$_4$Cl imaging buffer 180 s after stimulation. **a** Average sypHy response ($\Delta F/F_0 \pm$ SEM) normalised to the stimulation peak (stimulation indicated by bar, $n = 14$ DMSO, $n = 11$ 10 µM, $n = 10$ 30 µM). **b** Average level of sypHy fluorescence ($\Delta F/F_0 \pm$ SEM) at 224 s (One-way ANOVA, $n = 14$ DMSO, $n = 11$ 10 µM, $n = 10$ 30 µM, *$p = 0.0467$ DMSO vs 30 µM). **c** Peak level of sypHy fluorescence ($\Delta F/F_0 \pm$ SEM) normalised to the NH$_4$Cl challenge (One-way ANOVA, $n = 10$ DMSO, $n = 8$–10 µM, $n = 6$ 30 µM, all ns). **d–f** Primary cultures of hippocampal neurons prepared from either *Dnm1*[+/+] or *Dnm1*[+/R237W] embryos

were transfected with sypHy between 7 and 9 DIV. At 13–15 DIV, cultures were stimulated with a train of 300 action potentials (10 Hz) in the presence of either 30 µM BMS-204352 or a vehicle control (DMSO). Cultures were pulsed with NH$_4$Cl imaging buffer 180 s after stimulation (stimulation indicated by bar). **d** Average sypHy response ($\Delta F/F_0 \pm$ SEM) normalised to the stimulation peak ($n = 19$ DMSO *Dnm1*[+/+], $n = 15$ BMS *Dnm1*[+/+], $n = 16$ DMSO *Dnm1*[+/R237W], $n = 15$ BMS *Dnm1*[+/R237W]). **e** Average level of sypHy fluorescence ($\Delta F/F_0 \pm$ SEM) at 200 s (One-way ANOVA, $n = 19$ DMSO *Dnm1*[+/+], $n = 15$ BMS *Dnm1*[+/+], $n = 16$ DMSO *Dnm1*[+/R237W], $n = 15$ BMS *Dnm1*[+/R237W], *$p = 0.0183$ DMSO *Dnm1*[+/+] vs DMSO *Dnm1*[+/R237W]). **f** Peak level of sypHy fluorescence ($\Delta F/F_0 \pm$ SEM) normalised to the NH$_4$Cl challenge (One-way ANOVA $n = 19$ DMSO *Dnm1*[+/+], $n = 15$ BMS *Dnm1*[+/+], $n = 16$ DMSO *Dnm1*[+/R237W], $n = 15$ BMS *Dnm1*[+/R237W], all ns). Source data are provided as a Source Data file.

appears to be responsible for this phenotype, since homozygous *Fitful* mice display less SVs in inhibitory nerve terminals and overexpression of the A408T mutant inhibited receptor-mediated endocytosis in COS7 cells[36]. The absence of a dominant-negative effect of Dyn1$_{A408T}$-mCer on SV endocytosis in our study was therefore surprising. However, overexpression of dynamin-1 mutants in heterologous expression systems (where dynamin-2 is the dominant isoform) may result in more severe phenotypes when compared to the expression of these mutants in their natural context. This is supported by the relatively mild effect of both Dyn1$_{R237W}$-mCer and Dyn1$_{K44A}$-mCer on SV endocytosis in our study, which contrasts with the ablation of receptor-mediated endocytosis observed with the K44A mutant in non-neuronal heterologous expression systems[13].

One intriguing finding was the alteration in both spontaneous and evoked excitatory neurotransmission in *Dnm1*[+/R237W] mice with no parallel effect on inhibitory neurotransmission. The absence of effect on spontaneous or evoked inhibitory neurotransmission agrees with previous studies in the homozygous *Fitful* mouse (but see[37]), although increased rundown occurs during prolonged AP trains with a concomitant delay in the recovery of IPSC amplitude[16]. This relatively mild effect contrasts with studies in *Dnm1*[−/−] neurons, where there was a large reduction in evoked IPSCs when compared to EPSCs, and faster and more extensive depression of inhibitory neurotransmission during action potential trains[38]. Furthermore, in *Dnm1/3*[−/−] synapses, a strong facilitation of excitatory neurotransmission was observed during both low and high frequency stimulation, with mEPSC frequency, evoked

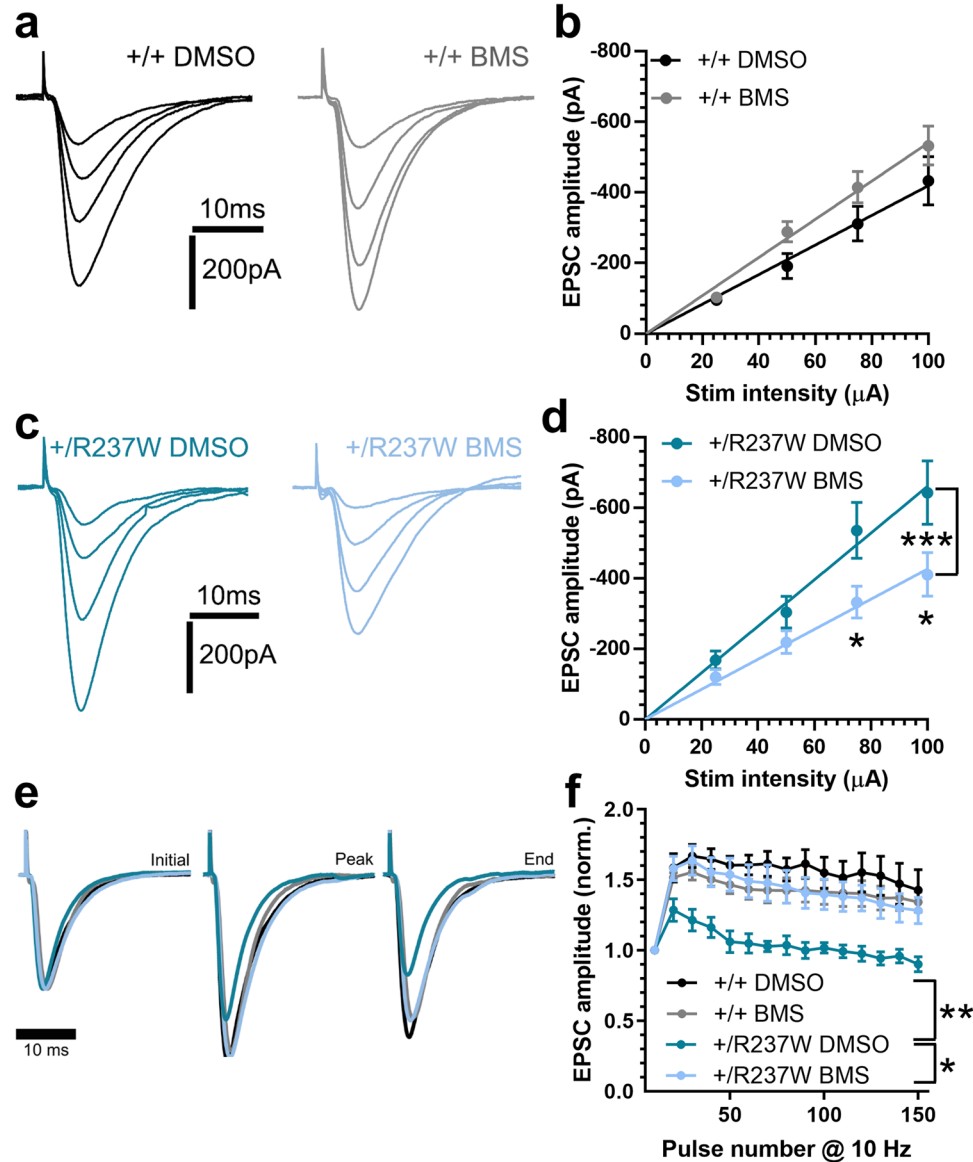

**Fig. 8 | BMS-204352 corrects neurotransmission defects in *Dnm1*[+/R237W] mice.**
Neurotransmission at CA3/CA1 synapses was monitored in acute hippocampal sli-
ces from either *Dnm1*[+/+] (**a**, **b**) or *Dnm1*[+/R237W] (**c**, **d**) mice. Slices were stimulated at a
range of intensities (25, 50, 75, and 100 µA, three repeats at each intensity, fre-
quency 0.05 Hz) in a pseudo random order in the presence of 30 µM BMS-204352
or a vehicle control (DMSO). Representative traces are displayed for either *Dnm1*[+/+]
(**a**) or *Dnm1*[+/R237W] (**c**) slices. **b, d** Evoked EPSC amplitude ± SEM is displayed (Two-
way ANOVA with Sidak's multiple comparison test, **b** *n* = 13 DMSO, *n* = 11 BMS, all ns;

**d** *n* = 13 DMSO, *n* = 13 BMS, ***\**p* = 0.0004, *\*p* = 0.041 75 µA, *\*p* = 0.0145 100 µA).
**e, f** *Dnm1*[+/+] or *Dnm1*[+/R237W] hippocampal slices stimulated with a 10 AP train (10 Hz),
in the presence of either 30 µM BMS-204352 or a vehicle control (DMSO).
**e** Representative traces, **f** evoked EPSC amplitude for normalised to first pulse ±
SEM. Two-way ANOVA with Dunnett's multiple comparison test, *n* = 5 DMSO
*Dnm1*[+/+], *n* = 6 BMS *Dnm1*[+/+], *n* = 5 DMSO *Dnm1*[+/R237W], *n* = 7 BMS *Dnm1*[+/R237W],
*\*\*p* = 0.002 DMSO *Dnm1*[+/+] vs DMSO *Dnm1*[+/R237W], *\*p* = 0.013 DMSO *Dnm1*[+/R237W] vs
BMS *Dnm1*[+/R237W]. Source data are provided as a Source Data file.

EPSC amplitude, RRP size and Pr all decreased[39]. *Dnm1*[+/R237W] synapses
also display reduced mEPSC frequency and decreased RRP, however in
contrast, we observe an increase in both Pr and evoked EPSCs in
addition to an absence of STP during action potential trains. Further-
more, we observe a decreased Pr at inhibitory *Dnm1*[+/R237W] synapses.
Therefore, even when SV endocytosis is disrupted to a similar extent
between *Dnm1*[+/R237W] and *Dnm1*[−/−] neurons, the dominant-negative
R237W mutation exerts discrete effects on circuit activity not observed
in models of loss of dynamin-1 function. Our observation of increased
Pr at excitatory synapses with a concomitant Pr decrease at inhibitory
synapses, therefore provides a potential microenvironment for epi-
leptogenesis, making it critical for future studies to determine how
*Dnm1*[+/R237W] neurons modify brain circuit properties and higher order
functions.

We observed a full correction of cellular, circuit and in vivo phe-
notypes via the delivery of BMS-204352. BMS-204352 was developed
as a BK channel agonist for the treatment of stroke[40], however in Phase
III trials it failed to display efficacy superior to placebo[25]. Nevertheless,
the drug exhibited an excellent safety profile, identifying it as a pro-
mising candidate for repurposing studies. Because BMS-204352 dis-
played both positive and negative modulatory effects against a
number of potassium channel subtypes[28,29], we employed a series of
channel openers and blockers to elucidate its mechanism of action.
Intriguingly, no drug from this palette of modulators recapitulated its
observed modifying activity on SV endocytosis. Therefore BMS-
204352 may have additional off-target effects responsible for its
reversal of phenotypes in the *Dnm1*[+/R237W] mouse. This question is
under active investigation.

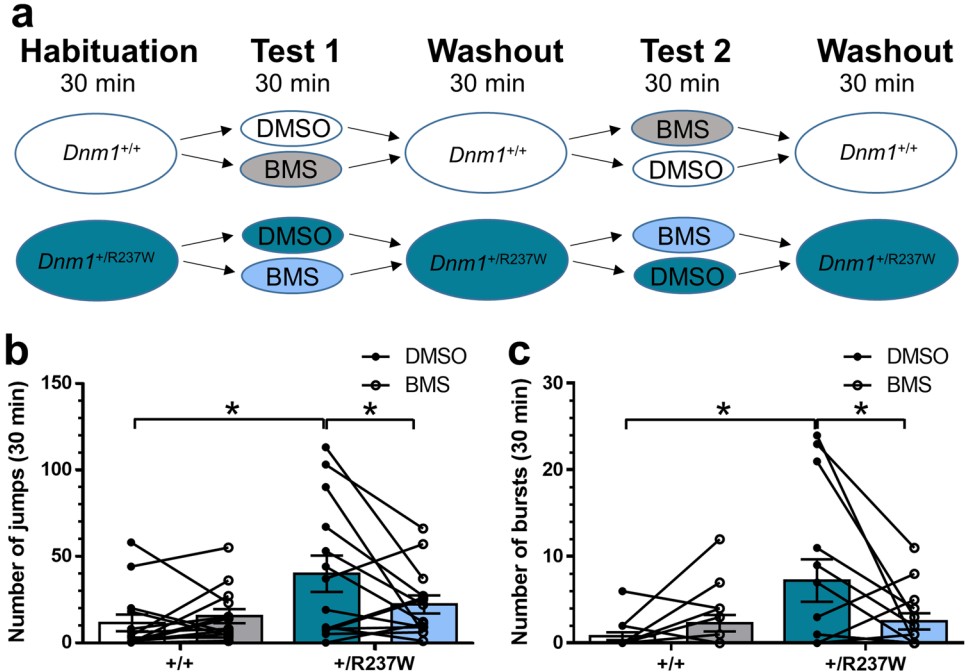

**Fig. 9 | BMS-204352 corrects seizure-like events in *Dnm1*[+/R237W] mice.** *Dnm1*[+/+] and *Dnm1*[+/R237W] mice were placed in an open field chamber for a 30 min period for 5 days. After habituation on day 1, mice were dosed with either BMS-204352 or a vehicle control (DMSO) on days 2 and 4, with drug washout on days 3 and 5. Delivery of drug treatment was interleaved between days 2 and 4. **a** Schematic of the experimental protocol. Average number of myoclonic jumps ± SEM (**b**) or bursts ± SEM (**c**).

**b** General linear model (repeated measures) with Bonferroni multiple comparisons $n = 14$ for all, *$p = 0.019$ DMSO *Dnm1*[+/R237W] vs BMS *Dnm1*[+/R237W], *$p = 0.021$ DMSO *Dnm1*[+/+] vs DMSO *Dnm1*[+/R237W], all other ns (**c**) General linear model (repeated measures) with Bonferroni multiple comparisons $n = 14$ for all, *$p = 0.016$ DMSO *Dnm1*[+/+] vs DMSO *Dnm1*[+/R237W] *$p = 0.011$ DMSO *Dnm1*[+/R237W] vs BMS *Dnm1*[+/R237W] all other ns. Source data are provided as a Source Data file.

The fact that BMS-204352 accelerates SV endocytosis and corrects cell, circuit and behavioural phenotypes in the *Dnm1*[+/R237W] mouse, provides strong evidence of a direct causal link between dysfunctional SV endocytosis and these outcomes. However, other interpretations are possible, since the R237W allele should perturb any dynamin-dependent endocytosis mode at the synapse. For example, altered trafficking of voltage-gated ion channels, and either pre- or post-synaptic receptors may also contribute to the observed phenotypes. *Dnm1*[+/R237W] excitatory synapses display no significant alterations in excitability in terms of action potential threshold, suggesting an absence of the intrinsic hyperexcitability observed in other NDDs models[26]. Action potential broadening and a decrease in action potential decay rate were observed however, which may contribute towards the enhancement in evoked EPSCs. Intriguingly, BK channels perform a key role in sculpting action potential shape[41]. However, BMS-204352 is not acting via this mechanism to correct function, since the observed broadening reflects reduced, rather than enhanced BK channel function. An increase in mIPSC amplitude is also observed, similar to the *Dnm1*[−/−] mouse[38]. This may reflect a compensatory recruitment of postsynaptic GABA_A receptors to offset increased excitability. Conversely, it may be a consequence of increased quantal release, due to larger SVs being formed via dysfunctional endocytosis[16]. The unaltered mEPSC amplitude in *Dnm1*[+/R237W] CA1 neurons suggests that postsynaptic stranding of AMPA receptors does not contribute to the increased evoked EPSC response however. One potential explanation for the increase in evoked EPSCs is the dysregulated retrieval of modulatory presynaptic receptors. A number of these have direct effects on Pr, via regulation of either ion channels or signalling cascades[42,43]. In support, we observed an upregulation of ErbB signalling molecules (which have direct effects on synaptic function[44]) in *Dnm1*[+/R237W] synaptosomes and a concomitant decrease in metabotropic receptor signalling molecules. It will be important to determine which of these signalling cascades directly contribute to

increased excitatory Pr and decreased inhibitory Pr, and which are compensatory changes that adjust for circuit hyperexcitability.

BMS-204352 is therefore a promising lead compound for future trials in *DNM1* epileptic encephalopathy. There is also potential for its use to be wider than this specific condition, since a cohort of monogenic neurodevelopmental disorders are predicted to have SV endocytosis defects at their core. For example, a series of frameshift, nonsense and missense mutations in essential SV endocytosis and SV cargo clustering genes have been identified in individuals with intellectual disability, autism and epilepsy[24], including the coat protein clathrin[45], adaptor protein complexes[46], SV cargo retrieval proteins[47–49] and regulators of endocytosis such as TBC1D24[50,51]. Furthermore, neurons derived from preclinical models for prevalent monogenic conditions such as fragile X syndrome and CDKL5 deficiency disorder have recently been discovered to display defects in SV retrieval[18,52]. With dysfunctional SV endocytosis emerging as a key convergence point in these monogenic conditions, upregulation of SV endocytosis via BMS-204352 may therefore provide a potential treatment to restore essential recycling mechanisms and normal function.

In conclusion, we have key cell, circuit and behavioural defects in a mouse model of *DNM1* epileptic encephalopathy, which provide important information on the molecular locus of seizure activity. Furthermore, an agent that accelerates SV endocytosis corrects all of these defects, suggesting intervention via this trafficking pathway is a promising therapeutic route.

## Methods

### Materials

Unless otherwise specified, all cell culture reagents were obtained from Invitrogen (Paisley, UK). Foetal bovine serum was from Biosera (Nuaille, France). Papain was obtained from Worthington Biochemical (Lakewood, NJ, USA). Caesium-gluconate, tetrodotoxin (TTX) and picrotoxin were from Hello Bio (Bristol, UK). Na$_2$GTP was from

Scientific Laboratory Supplies (Newhouse, UK), whereas Na$_2$-creatine was from (Merck, London, UK). BMS-204352 was from Bio-Techne Ltd (Abingdon, UK). All other reagents were obtained from Sigma-Aldrich (Poole, UK) unless specified. Synaptophysin-pHluorin (sypHy) was provided by Prof. L. Lagnado (University of Sussex, UK). Rat dynamin-1aa fused to mCerulean at its C-terminus[53] was subjected to site-directed mutagenesis to generate both R237W (forward primer ATTGGCGTGGTGAACTGGAGCCAGAAGGACATA, reverse primer TAT GTCCTTCTGGCTCCAGTTCACCACGCCAAT) and K44A mutations (forward primer GGCCAGAGCGCCGGCGCGAGCTCGGTGCTGGAC, reverse primer CTCCAGCACCGAGCTCGCGCCGGCGCTCTGGCC). Base changes were confirmed by Source Bioscience Sanger Sequencing (Glasgow, UK).

## Generation of *Dnm1*$^{+/R237W}$ mice

The *Dnm1*$^{+/R237W}$ mouse was generated by Horizon Discovery (St. Louis, USA). Briefly, the codon encoding R237 within the *Dnm1* gene was targeted using CRISPR-Cas9 technologies on a C57Bl/6J genetic background using the guide sequence cgtggtgaaccggagccagaagg. This resulted in the modification of the *Dnm1* gene sequence from CGGAGC (equivalent amino acids 237/238−RS) to TGGTCT (amino acids 237/238−WS). In total, 44 animals were screened for the point mutation, with 4 found to be positive. Two founders were backcrossed to *Dnm1*$^{+/+}$ mice to generate F1 heterozygous progeny. The F1 progeny of one of the founder lines was taken forward to establish the colony. Mice were maintained as heterozygotes by crossing *Dnm1*$^{+/R237W}$ mice with C57Bl/6J *Dnm1*$^{+/+}$ mice, with three backcrosses every five generations. Genotyping was performed by Transnetyx (Cordova, TN, USA). A separate in-house colony of C57Bl/6J *Dnm1*$^{+/+}$ mice were used as a source of tissue for hippocampal cultures in experiments where dynamin-1 variants were overexpressed.

Animal work was performed in accordance with the UK Animal (Scientific Procedures) Act 1986, under Project and Personal Licence authority and was approved by the Animal Welfare and Ethical Review Body at the University of Edinburgh (Home Office project licences – 7008878 and PP5745138 to Prof. Cousin and PP1538548 to Dr. Gonzalez-Sulser). Specifically, all animals were killed by Schedule 1 procedures in accordance with UK Home Office guidelines; adults were killed by cervical dislocation or exposure to CO$_2$ followed by decapitation, whereas embryos were killed by decapitation followed by destruction of the brain. The in-house colony of C57Bl/6J *Dnm1*$^{+/+}$ mice and the *Dnm1*$^{+/R237W}$ mouse colony were housed in standard open top caging on a 14/10 h light/dark cycle (light 7 A.M. to 9 P.M.). Breeders were fed RM1 chow, whereas stock mice were maintained on RM3 chow. The ambient temperature ranged between 19 and 23 °C with humidity 55 ± 10%.

## Cell culture and transfections

Heterozygous *Dnm1*$^{+/R237W}$ mice were mated with *Dnm1*$^{+/+}$ mice to produce either *Dnm1*$^{+/+}$ or *Dnm1*$^{+/R237W}$ offspring. Hippocampi from each embryo were processed separately to avoid contamination across genotypes. Dissociated primary hippocampal cultures were prepared from embryos as previously described[18]. Briefly, isolated hippocampi were digested in a 10 U/mL papain solution (Worthington Biochemical, LK003178) at 37 °C for 20 min. The papain was then neutralised using DMEM F12 (ThermoFisher Scientific, 21331-020) supplemented with 10% foetal bovine serum (BioSera, S1810-500) and 1% penicillin/streptomycin (ThermoFisher Scientific, 15140-122). Cells were triturated to form a single cell suspension and plated at $5 \times 10^4$ cells per coverslip on laminin (10 µg/mL; Sigma Aldrich, L2020) and poly-D-lysine (Sigma Aldrich, P7886) coated 25 mm glass coverslips (VWR International Ltd, Lutterworth, UK). Cultures were maintained in Neurobasal media (ThermoFisher Scientific, 21103-049) supplemented with 2% B-27 (ThermoFisher Scientific, 17504-044), 0.5 mM L-glutamine (Thermo-Fisher Scientific, 25030-024) and 1% penicillin/streptomycin. After

2–3 days in vitro (DIV), 1 µM of cytosine arabinofuranoside (Sigma Aldrich, C1768) was added to each well to inhibit glial proliferation. Hippocampal neurons were transfected with synaptophysin-pHluorin (sypHy) and/or Dyn1-mCer using Lipofectamine 2000 (ThermoFisher Scientific, 11668027) as per manufacturer's instructions and imaged at DIV 13–15.

## Imaging of SV recycling using sypHy

Imaging of SV recycling was monitored using sypHy as previously described[18]. SypHy-transfected hippocampal cultures were mounted in a Warner Instruments (Hamden, CT, USA) imaging chamber with embedded parallel platinum wires (RC-21BRFS) and were mounted on a Zeiss Axio Observer D1 inverted epifluorescence microscope (Cambridge, UK). Neurons were challenged with field stimulation using a Digitimer LTD MultiStim system-D330 stimulator (current output 100 mA, current width 1 ms) either at 10 Hz for 30 s or 40 Hz for 10 s. Neurons were visualised at 500 nm band pass excitation with a 515 nm dichroic filter and a long-pass >520 nm emission filter, with images captured using an AxioCam 506 mono camera (Zeiss) with a Zeiss EC Plan Neofluar 40×/1.30 oil immersion objective. Image acquisition was controlled using Zen Pro software (Zeiss). Imaging time courses were acquired at 4 s intervals while undergoing constant perfusion with imaging buffer (119 mM NaCl, 2.5 mM KCl, 2 mM CaCl$_2$, 2 mM MgCl$_2$, 25 mM HEPES, 30 mM glucose at pH 7.4, supplemented with 10 µM 6-cyano-7-nitroquinoxaline-2,3-dione (CNQX, Abcam, Cambridge, UK, ab120271) and 50 µM DL-2-Amino-5-phosphono-pentanoic acid (AP5, Abcam, Cambridge, UK, ab120044). Alkaline buffer (50 mM NH$_4$Cl substituted for 50 mM NaCl) was used to reveal the maximal pHluorin response. SV fusion during stimulation was measured by stimulating sypHy-transfected neurons (10 Hz, 90 s) in the presence of 1 µM bafilomycin A1 (Cayman Chemical Company, Ann Arbor Michigan, USA, 11038). BMS-204352 and other potassium channel modulators in imaging buffer were perfused over neurons 2 min prior to and during imaging until addition of alkaline buffer.

Time traces were analysed using the FIJI distribution of Image J (National Institutes of Health). Images were aligned using the Rigid body model of the StackReg plugin (https://imagej.net/StackReg). Nerve terminal fluorescence was measured using the Time Series Analyser plugin (https://imagej.nih.gov/ij/plugins/time-series.html). Regions of interest (ROIs) 5 pixels in diameter were placed over nerve terminals that responded to the electrical stimulus. A response trace was calculated for each cell by averaging the individual traces from each selected ROI. Inhibition of SV endocytosis was calculated as remaining fluorescence 140 s after termination of stimulation. The time constant of SV endocytosis could not be calculated, since individual sypHy traces within each set of experiments did not conform to first order kinetics.

## HRP uptake in hippocampal cultures

Hippocampal cultures were mounted in the RC-21BRFS stimulation chamber and challenged with 400 action potentials (40 Hz) in the presence of 10 mg/ml HRP (Sigma Aldrich, P8250) supplemented imaging buffer. Immediately following the end of stimulation, cultures were washed in imaging buffer to remove non-internalised HRP and fixed with a solution of 2% glutaraldehyde (Electron Microscopy Sciences, Hatfield, USA, 16019) and 2% PFA in 0.1 M in phosphate buffer (PB). After washing in 0.1 M PB, HRP was developed with 0.1% 3,3′-diaminobenzidine (Fluka Chemica, Gillingham, UK, 22204001) and 0.2% v/v hydrogen peroxide (Honeywell, Muskegon, USA, 216763) in PB. After further washing in PB, cultures were stained with 1% osmium tetroxide (TAAB laboratory and microscopy, Aldermaston, UK, O015/1) for 30 min. Samples were then dehydrated using an ethanol series and polypropylene oxide (Electron Microscopy Sciences, Hatfield, USA, 20411) and embedded using Durcupan resin (Sigma Aldrich, 44610). Samples were sectioned, mounted on grids, and viewed using an FEI

Tecnai 12 transmission electron microscope (Oregon, USA). Intracellular structures that were <61 nm in diameter were arbitrarily designated to be SVs, whereas larger structures were considered endosomes. The area of individual endosomes was obtained by tracing the circumference using the freehand selections tool in ImageJ and measuring the resulting area. Typically, 20 fields of view were acquired for one coverslip of cells. In nerve terminals that contained HRP, the average number of HRP-labelled endosomes and SVs per nerve terminal was calculated for each coverslip and represents the experimental $n$.

## Quantification of endocytic profiles

Two-month-old $Dnm1^{+/+}$ or $Dnm1^{+/R237W}$ mice were terminally anaesthetised by intraperitoneal overdose of sodium pentobarbital. Mice were then perfused through the left ventricle with ice-cold 0.1 M phosphate buffer, followed by a fixative solution consisting of 2% paraformaldehyde and 2% glutaraldehyde in 0.1 M phosphate buffer. Brains were dissected and post-fixed in the same fixative solution overnight at 2–4 °C, at which point the fixative solution was replaced with 0.1 M phosphate buffer. Brains were then processed for electron microscopy as described above. Individual endosomes and SVs were counted and normalised to the nerve terminal area. The area of individual endosomes and SVs was obtained by fitting a region of interest over each individual structure using the round area selection tool in ImageJ and measuring the resulting area. Intracellular structures that were <2922 nm$^2$ in area were arbitrarily designated to be SVs, whereas larger structures were considered endosomes.

## Immunocytochemistry

Immunofluorescence staining and analysis was performed as previously described[18]. Briefly, hippocampal neurons were fixed with 4% paraformaldehyde (PFA) in PBS for 15 min at room temperature. PFA was then removed and cells were quenched 2 × 5 min with 50 mM NH$_4$Cl in PBS. Cells were then washed 4 × 5 min with PBS. Before staining, cells were permeabilised in 1% bovine serum albumin (BSA) in PBS-Triton 1% for 5 min. Cells were then washed in PBS before blocking in 1% BSA in PBS at room temperature for 1 h. After blocking, cells were left to incubate in primary antibody diluted in blocking solution for 30–45 min (chicken anti-GFP (Abcam ab13970) 1:500; rabbit anti-SV2A (Abcam ab32942) 1:200; goat anti-dynamin-1 (Santa Cruz sc-6402) 1:200). Following 4 × 5 min washes, cells were left to incubate in secondary antibody (goat anti-chicken Alexa-Fluor-488 (Invitrogen A11039) 1:1000; goat anti-rabbit Alexa-Fluor-568 (Invitrogen A21069) 1:1000; donkey anti-goat Alexa-Fluor-647 (Invitrogen A21447) 1:1000) diluted in blocking buffer for 30-45 min at room temperature in the dark. After washing, coverslips were mounted to slides using FluorSave Reagent (Millipore). Alexa Fluor 488 and 568 images were acquired using a dual camera imaging system (Zeiss). The signal was filtered by a double band pass excitation filter (470/27 + 556/25) with beam splitter (490 + 575) and emission filters 512/30 and 630/98 (Zeiss) respectively. Alexa Fluor 647 was visualised with a 640 nm excitation and a 690/50 band pass emission filter. For each image analysed, ROIs were placed over the transfected neuron, a non-transfected neuron and the background. This allowed measurement of levels of overexpression of mCer-Dyn1 within neurons on the same coverslip by comparing the overexpression to normal expression levels. Background fluorescence was subtracted from all signals. For each coverslip, 4–6 fields with transfected neurons were acquired. The n is the number of transfected cells imaged.

## Immunohistochemistry

Two-month-old $Dnm1^{+/+}$ or $Dnm1^{+/R237W}$ littermate male mice were administered a lethal dose of sodium pentobarbital and transcardially perfused with cold PBS followed by cold PFA (PFA, 4% in 0.1 M PB). Brains were extracted and fixed for 24 h in PFA at 4 °C, washed with PBS, and transferred to a 30% sucrose / PBS solution for 48 h at 4 °C. Brains were embedded in tissue freezing compound and 50 μm coronal sections were generated using a freezing microtome. Free floating thin sections were permeablised for 4–5 h in block solution (PBS, 10% horse serum, 0.5% BSA, 0.5% Triton X-100, 0.2 M glycine) then incubated with NeuN primary antibody diluted in block solution (1:1000; Merck; Cat # MAB377) overnight at 4 °C. Slices were washed 4–5 times in PBS for 2 h then incubated for 3-4 h with secondary antibody (anti-rabbit Alexa Fluor 568; 1:1000; Invitrogen; Cat #A10042) and Neuro-Trace Green Fluorescent Nissl Stain (1:2000; Invitrogen; Cat #N21480) at room temperature. Slices were then washed 4–5 times in PBS for 2 h and mounted onto glass slides using ProLong Gold Antifade Mountant (Invitrogen; Cat #P36930). Sections were imaged on a Leica SP8 upright confocal laser scanning microscope using a ×10/NA 0.45 objective. The tile function within the Leica software was used to acquire overlapping images over the whole section followed by the merge image processing function to stitch the tiles together.

## Protein biochemistry

Cultured hippocampal neurons from $Dnm1^{+/+}$ or $Dnm1^{+/R237W}$ sex-matched littermates at DIV 14 were lysed directly into SDS (sodium dodecylsulfate) sample buffer (67 mM Tris, pH 7.4, 2 mM EGTA, 9.3% glycerol, 12% β-mercaptoethanol, bromophenol blue, 67 mM SDS) and boiled at 95 °C for 10 min prior to Western blotting. Whole brain lysates were prepared from the brains of 3 week old and 6 week old age-matched $Dnm1^{+/+}$ or $Dnm1^{+/R237W}$ mice. Brain homogenates were prepared in RIPA buffer (10 mM Tris-HCl, pH 8.0, 1 mM EDTA, 0.5 mM EGTA, 15% Triton X-100, 0.1% sodium deoxycholate, 0.1% SDS, 140 mM NaCl, and 1 mM PMSF) and centrifuged in a Beckman-Coulter Optima-Max ultracentrifuge at 116,444 $g$ for 40 min at 4 °C. Protein concentration was determined using a Bradford (Applichem, Germany; A6932) assay following manufacturer's instructions. SDS sample buffer was added to the lysates and samples were boiled for 10 min before loading on SDS-PAGE and transfer onto nitrocellulose membranes. Membranes were incubated with primary antibodies overnight at 4 °C (Goat anti-amphyphysin-1 (Santa Cruz sc-8536) 1:500; rabbit anti-Eps15 (Santa Cruz sc-534) 1:1000; Goat anti-dynamin-1 (Santa Cruz sc-6402) 1:1000; mouse anti-synaptotagmin-1 (Abcam ab13259) 1:500; rabbit anti-syndapin-1 (Abcam ab137390) 1:4000; goat anti-endophilin-A1 (Santa Cruz sc-10874) 1:1000; rabbit anti-C-src (Santa Cruz sc-19) 1:100; mouse anti-actin (Sigma Aldrich A4325) 1:50000). Secondary antibodies were incubated for 1 h at room temperature (all Li-Cor, 1:10000; donkey anti-goat (IRDye® 680RD, 926-68074); donkey anti-goat (IRDye® 800CW, 926-32214); donkey anti-rabbit (IRDye® 800CW, 926-32213); donkey anti-mouse (IRDye® 800CW, 926-32212); goat anti-mouse (IRDye® 680RD, 926-68070). Membranes were imaged on an Odyssey 9120 Infrared Imaging System (LI-COR Biosciences) using *LI-COR* Image Studio Lite software (version 5.2) and analysed using ImageJ. The integrated density of signals was measured in rectangular ROIs of an identical size set around the protein expression bands.

## Mass spectrometry

Synaptosomes were prepared from two-month-old $Dnm1^{+/+}$ or $Dnm1^{+/R237W}$ littermate male mice as described[30]. Briefly, animals were culled by cervical dislocation with death confirmed by destruction of the brain via homogenisation in ice-cold 0.32 M sucrose, 5 mM EDTA (pH 7.4) after removal of the cerebellum. The homogenate was centrifuged at 950 × $g$ for 10 min at 4 °C at which point the supernatant was saved and the pellet was resuspended in the same sucrose buffer. The resuspended pellet solution was centrifuged at 950 × $g$ for 10 min at 4 °C and the resulting supernatant was combined with the first. The combined supernatant was then centrifuged at 20,400 × $g$ for 30 min at 4 °C and the pellet (crude synaptosomal fraction) was retained. Synaptosome pellets were dissolved in Urea lysis buffer (8 M Urea in 50 mM Tris-Cl and 1% sodium deoxycholate) and were quantified using

the BCA method. 20 µg of total protein was used for proteomic sample preparation by suspension trapping (S-Trap)[54], as recommended by the supplier (ProtiFi, Huntington NY, USA). Samples were reduced with 5 mM Tris (2-carboxyethyl)phosphine (Pierce) for 30 min at 37 °C, and subsequently alkylated with 5 mM IAM (Iodoacetamide) for 30 min at 37 °C in the dark. After acidification with phosphoric acid, sample was cleaned and digested using Trypsin (1:20) as mentioned by in the manufacturer's protocol using S-trap filter for 2 h at 47 °C and the digested peptides are eluted using 0.2% Formic acid and 50% Acetonitrile: 0.2% formic acid. The eluted digested peptides were dried in speed vac and stored at −80 °C.

The peptides were reconstituted in 30 µL of 0.1% formic acid and vortexed and 5 µL of each sample was injected on the mass spectrometer. Peptides were analysed by nanoflow-LC-MS/MS using a Orbitrap Q-Exactive-HF™ Mass Spectrometer (Thermo Scientific ™) coupled to a Dionex™ Ultimate™ 3000. Samples were injected on a 100 µm ID × 5 mm trap (Thermo Trap Cartridge 5 mm) and separated on a 75 µm × 50 cm nano LC column (EASY-Spray™ LC Columns #ES803). All solvents used were HPLC or LC-MS Grade (Millipore™). Peptides were loaded for 5 min at 10 µL/min using 0.1% FA, 2% Acetonitrile in Water. The column was conditioned using 100% Buffer A (0.1% FA, 3% DMSO in Water) and the separation was performed on a linear gradient from 0 to 35% Buffer B (0.1% FA, 3% DMSO, 20% Water in Acetonitrile), over 140 min at 250 nL/min. The column was then washed with 90% Buffer B for 5 min and equilibrated 10 min with 100% Buffer A in preparation for the next analysis. Full MS scans were acquired from 350 to 1500 $m/z$ at resolution 60,000 at $m/z$ 200, with a target AGC of $3 \times 10^6$ and a maximum injection time of 50 ms. MS/MS scans were acquired in HCD mode with a normalised collision energy of 25 and resolution 15,000 using a Top 20 method, with a target AGC of $2 \times 10^5$ and a maximum injection time of 50 ms. The MS/MS triggering threshold was set at 5E3 and the dynamic exclusion of previously acquired precursor was enabled for 45 s for DDA mode. For DIA mode the scan range was 385 to 1015 $m/z$, where MS/MS data were acquired in 24 m/z isolation windows at a resolution of 30,000.

Pooled peptides from all samples were fractionated on a Basic Reverse Phase column (Gemini C18, 3 µm particle size, 110 A pore, 3 mm internal diameter, 250 mm length, Phenomenex #00G-4439-Y0) on a Dionex Ultimate 3000 Off-line LC system. All solvent used were HPLC grade (Fluka). Peptides were loaded on column for 1 min at 250 µL/min using 99% Buffer A (20 mM Ammonium Formate, pH = 8) and eluted for 48 min on a linear gradient from 2 to 50% Buffer B (100% ACN). The column is then washed with 90% Buffer B for 5 min and equilibrated for 5 min for the next injection. Peptide elution was monitored by UV detection using at 214 nm. Fractions were collected every 45 s from 2 min to 60 min for a total of 12 fractions. Nonconsecutive concatenation of every 13th fraction was used to obtain 12 pooled fractions (Pooled Fraction 1: Fraction 1 + 13 + 25 + 37, Pooled Fraction 2: Fraction 2 + 14 + 26 + 38...).

## Data analysis

Label-free quantitative analysis was performed using the data set acquired in DIA mode. Peptide identification was carried out using a library generated using both DDA and DIA datasets using Spectronaut™ version 15.0. The library was generated using the Pulsar algorithm integrated in Spectronaut using Mus musculus FASTA using 1% FDR. The maximum of missed cleavage was set to 2 using Trypsin/P enzyme. Carbamidomethylation (C) was set as fixed modification and acetylation (Protein N term), oxidation (M), deamination (NQ), were set as variable modifications. The library consisted spectra information of 5906 proteins in total. DIA data set for both WT and HET was searched using this library quantified 4237 proteins in total. Statistical analysis was done using R script and limma package was used for making contrasts. Raw proteomic data were deposited on PRIDE (https://www.ebi.ac.uk/pride/) as outlined below.

## Analysis of mass spectrometry data

Gene Ontology terms enrichment analysis on the upregulated and downregulated proteins was performed against Mus musculus background using Database for Annotation, Visualization and Integrated Discovery (DAVID). Detailed enrichment analysis are available in Supplementary Data 2. Network analysis of the upregulated and downregulated proteins were performed using the STRING web tool (v.11.5).

Enrichment analysis of the full protein list were performed using ShinyGO v0.76.2 for the Cellular Component and Biological Pathways, selected by FDR and sorted by FoldEnrichment and using the synapse specific database SynGO[55] against the 'brain expressed' background, setting medium stringency and second level terms as labels for Cellular Component representation and top levels terms as labels for Biological Pathways representation (Supplementary Fig. 3).

## GTPase assays

A colorimetric assay was used to quantify GTPase activity of the different mCer-Dyn1 mutants[56]. HEK293T cells transfected with mCer-Dyn1 plasmids were harvested 48 h after transfection with a 1:1 ratio of Lipofectamine2000 to plasmid. The cells were resuspended in 1 ml of sucrose lysis buffer (250 mM sucrose, 3 mM imidazole pH 7.4 supplemented with 2 µl/ml protease inhibitors and 1 mM phenylmethane sulfonyl fluoride) and mechanically broken using a primed ball-bearing cell cracker (EMBL, Heidelberg, Germany). Anti-GFP VHH coupled to agarose beads for immunoprecipitation of GFP-fusion proteins (GFP-Trap; ChromoTek GmbH, Germany; gta-20) was used for immunoprecipitation of mCer, mCer-Dyn1WT or mutant mCer-Dyn1 according to manufacturer's instructions. A Bradford (Applichem, Germany; A6932) assay was performed according to manufacturer's instructions to determine protein concentration of GFP-Trap-bound mCer or mCer-Dyn1. GFP-Trap-bound mCer-Dyn1 mutants were diluted to a concentration of 1 µM in GTPase assay buffer (20 mM HEPES pH 7.5, 50 mM KCl−this low salt concentration allows for the oligomerisation of dynamin[57], 2 mM MgCl₂). For each reaction, 20 µl of 2 mM GTP diluted in GTPase assay buffer and 20 µl of 1 µM stock of mCer-Dyn1 was incubated for 30 min at 37 °C after which 0.5 M EDTA pH 8.0 was added to terminate the reaction. 300 µl of filtered Malachite green solution (34 mg Malachite green carbinol base dissolved in 40 ml of 1 N HCl added to 1 g of ammonium molybdate tetrahydrate diluted in 14 mL of 4 N HCl up to 100 mL with ddH₂O) was added to each reaction. The change in colour of malachite green was quantified using a plate reader to measure the absorbance at 650 nm. The amount of inorganic phosphate released was calculated using the standard curve.

## Acute slice preparation

Horizontal hippocampal slices (350 µm) were prepared from *Dnm1*^+/R237W^ and *Dnm1*^+/+^ littermate control mice (P19-25 of either sex). Animals were culled by cervical dislocation with death confirmed by removal of the brain. Excised brains were rapidly transferred to chilled (2−5 °C) carbogenated sucrose-modified artificial cerebrospinal fluid (saCSF in mM: NaCl 86, NaH₂PO₄ 1.2, KCl 2.5, NaHCO₃ 25, glucose 25, sucrose 50, CaCl₂ 0.5, and MgCl₂ 7) for 2 min and subsequently sliced in the same solution using a vibrating microtome (Leica VT1200S). Slices were allowed to recover for 1 h at 33 °C in carbongenated standard aCSF which contained (mM): NaCl 126, KCl 3, NaH₂PO₄ 1.2, NaHCO₃ 25, glucose 15, CaCl₂ 2, and MgCl₂ 2.

## Electrophysiology

For recording, slices were transferred to an immersion chamber continuously perfused with standard aCSF (MgCl₂ 1 mM) maintained at 32 °C using an in-line Peltier heater (Scientifica, Uckfield, UK). A cut was made between CA2 and CA1 (identified as the medial termination of stratum lucidum) to ablate recurrent activity. Whole-cell patch-clamp recordings were made from visually identified pyramidal neurons in

the CA1 region using pulled borosilciate electrodes (4–7 MΩ). The intracellular solution for evoked and intrinsic properties experiments consisted of (mM): K-gluconate 142, KCl 4, EGTA 0.5, HEPES 10, MgCl$_2$ 2, Na$_2$ATP 2, Na$_2$GTP 0.3, and Na$_2$-creatine 10. For mEPSC recordings, a caesium-based intracellular solution was used (mM): Cs-gluconate 140, CsCl 3, EGTA 0.2, HEPES 10, QX-314 chloride 5, MgATP 2, NaATP 2, Na$_2$GTP 0.3, and phosphocreatine 10. Excitatory currents were recorded in the presence of picrotoxin (50 μM) with cells voltage-clamped at −70 mV, inhibitory currents were recorded in the presence of CNQX (10 μM) and D-AP5 (50 μM) with cells voltage-clamped at −10 mV. A further addition of TTX (300 nM) was made for mPSC recording. For experiments with BMS-204352, the drug was dissolved in DMSO and TWEEN® 80 before adding to standard aCSF. Final drug concentration was 30 μM, with the vehicles both at 0.03% v/v.

Recording protocols: Intrinsic properties were recorded in current-clamp mode. All other recordings were made under voltage-clamp. Currents were low pass filtered at 3–10 kHz and sampled at 10–20 kHz, using Clampex 10 software (pClamp 10, Molecular Devices, San Jose, USA). For evoked recordings, Schaffer collaterals were stimulated with a patch electrode (-1–2 MΩ) filled with aCSF and positioned in stratum radiatum, connected to an isolated constant current stimulator (Digitimer, Welwyn Garden City, UK). In all cases, the stimulus intensity was set to evoke a current of -200 pA following a 50 μs pulse. Stimulus was delivered at either: paired pulses (interval 10–500 ms, pairs 30 s apart), or long trains (either 10 or 40 Hz for 15 s, four repeats delivered 4 min apart). Data were analysed offline using either the open source Stimfit software package (intrinsic properties) or Clampfit from the pClamp 10 software suite (all EPSCs). Cells were excluded from analysis if series resistance varied by more than 20% during recording.

RRP size and its replenishment were determined using approaches described in ref. [23]. Briefly RRP was calculated by plotting the cumulative EPSC amplitude from 40 Hz 15 s trains, and performing a linear regression on the last 1 s of that plot. The y-intercept of this regression line denotes RRP size (Fig. 5g). Replenishment rate is represented by the slope of the regression line. Pr was calculated as amplitude of the first evoked EPSC divided by the effective RRP size[58].

For quantification of IPSC PPR, the amplitude of the second response was measured from the lowest point immediately following the second stimulation artefact. This was because the typical decay kinetics of inhibitory responses meant that the response had not returned to baseline prior to the onset of the second stimulus.

## Surgery for in vivo electrophysiology

*Dnm1*[+/R237W] and *Dnm1*[+/+] littermate control mice of either sex aged 8 weeks were anaesthetised with isoflurane and mounted on a stereotaxic frame (David Kopf Instruments, USA). Pairs of local LFP electrodes (Ø = 50.8 μm, Teflon insulated stainless steel, A-M Systems, USA) were implanted targeting dorsal hippocampus bilaterally (1.85 mm caudal, 1.25 mm lateral from bregma and 1.40 mm ventral from brain surface), ventral hippocampus bilaterally (3.3 mm caudal, 3.3 mm lateral from bregma and 2.9 mm ventral from the brain surface), left motor cortex (1.55 mm caudal, 1.88 mm left from bregma and on the brain surface), right somatosensory cortex (1.3 mm caudal, 2.0 mm lateral from bregma and on the brain surface) and the midline cerebellum (5.7 mm caudal, 0 mm lateral from bregma and on the brain surface). Two miniature ground screws (Yahata Neji, M1 Pan Head Stainless Steel Cross, RS Components, Northants, UK) were attached over the cerebellum (5.0 mm caudal, 2 mm lateral) to serve as ground as well as three additional screws for structural support. The electrodes were attached to an electronic interface board (EIB-16, Neuralynx, USA). The electrode assemblies were fixed to the skull using a combination of UV activated cement (3M Relyx Unicem 2 Automix, Henry Schein, Gillingham, UK) and dental cement (Simplex Rapid, Kemdent, Swindon, UK).

## In vivo LFP recordings

Mice were placed in 50 × 50 cm square arenas and connected for recordings to an RHD 16-channel recording headstage (Intantech, USA) through an electrical commutator (Adafruit, USA) and an acquisition board (Open Ephys, USA). LFP signals were sampled at 1 kHz and referenced to ground using OpenEphys GUI (Open Ephys, USA). Mice were video-recorded during stimulation sessions at 9.98 frames/s (C270 HD webcam, Logitech, USA). A 1 s light pulse from a blue LED (blue = 465 nm, Plexon, USA) mounted on each commutator was triggered by a Master-8 (AMPI) every five min to synchronise jump timestamps in video and LFP recordings.

## Analysis of in vivo LFP recordings

Jump timestamps were identified by visual analysis of concurrent videos in 1 h recordings. Between 2 and 8 jumps per animal were analysed and values were averaged per mouse. The power spectrum, with a 1 s non-overlapping Hann window, was calculated from the dorsal hippocampus for 1 s after the start of a jump, using the SciPy Python function Periodogram. The duration of electrophysiological activity was manually measured by plotting the data with the plot function from the MNE Python package.

## Behavioural experiments

For the open field assay, 6- to 8-week-old *Dnm1*[+/R237W] and *Dnm1*[+/+] littermate controls of either sex were placed in an open field arena 50 cm × 50 cm for 30 min for 5 consecutive days. The first day in the arena served as habituation. On days 2 (test 1) and 4 (test 2) mice received 2 mg/kg BMS-204352 or vehicle (DMSO 1/80; Tween 80 1/80; 0.9% NaCl) administered by intraperitoneal injection (as described in ref. [26]) in a counterbalanced manner as described in Fig. 9a. No injections were administered on days 3 and 5 (washout). Injections were administered 20 min prior to start of experiment to ensure maximal brain BMS-204352 concentration for duration of time in open field arena. Activity was recorded at 9.89 fps from both the top view and side view of the arena using Logitech cameras (C270 HD webcam, Logitech) with up to four animals being recorded simultaneously in individual arenas. Jumping behaviour was scored using Behavioral Observation Research Interactive Software (BORIS v.7.9.24, University of Torino[59]) which allowed for each jump to be logged in time.

## Analysis of mouse movement and location in behavioural tasks

DeepLabCut (DLC v.2.1.10.4) was used to compare the movement and position of mice[60]. The tail base was used for analysis, since it provided the most accurate approximation of movement and position in two-dimensions. DLC tracked the movement of the animal for the duration of each video and provided an output for its X- and Y-coordinates at every frame. A loop was used to iterate the predicted tail base coordinates in each video and calculate the distance an animal travelled between each frame. These distances were summed across frames to determine total distance covered in the 30 min experiment. Videos were then grouped based on their camera angle. For each angle, DLC was used to assign coordinates to the corners of the animal's arena, which allowed conversion of DLC units into centimetres. To determine the time an animal spent in the centre and along the walls of the arena, different camera angles were used. For each angle, the dimensions of the animal's arena were approximated to create an 'outer' and an 'inner' box. Each box contained half the total area of the arena. The number of tail base coordinates found within both boxes were totalled (time spent in centre), as well as those only found within the large box (time spent at edges).

## Statistical analysis

Experimenters were blinded to the genotype of both animals and cells for all experiments and data analysis. Statistical analysis was performed using GraphPad Prism 8.4.3. Statistical analysis for paired

behaviour data was analysed using IBM SPSS Statistics v29. No statistical methods were used to predetermine sample sizes and no randomisation procedures were applied. Statistical tests were applied based on the distribution of the datasets measured using D'Agostino-Pearson normality test. Significance was set at ns $P > 0.05$, $*P < 0.05$, $**P < 0.01$, $***P < 0.001$, $****P < 0.0001$. Mann-Whitney (two-tailed), Wilcoxon matched-pairs signed rank (two-tailed), and Kruskal-Wallis with Dunns post-hoc tests were used to compare non-Gaussian data sets. Student's t test (two-tailed) and analyses of variance followed by Dunnett's post-hoc test were used to compare normally distributed data sets. General linear model (repeated measures) was used to determine genotype effects, treatment effects and interactions. Bonferroni multiple comparisons test was used for multi-group comparisons where appropriate. Information about sample sizes, statistical tests used to calculate $P$ values and the numeric values of the results are specified in figure legends and Supplementary Table 2.

### Reporting summary

Further information on research design is available in the Nature Portfolio Reporting Summary linked to this article.

## Data availability

All relevant data are included in the article and/or its supplementary information files. Source data are provided within this paper. The one exception is the raw proteomic data, which is deposited on PRIDE (Project accession: PXD039667; Project title: Reversal of cell, circuit and seizure phenotypes in a mouse model of DNM1 epileptic encephalopathy; Project webpage: http://www.ebi.ac.uk/pride/archive/projects/PXD039667). Source data are provided with this paper.

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

## Acknowledgements

This research was funded by grant awards to M.A.C. (Epilepsy Research UK (P2003), Wellcome Trust Investigator Award (204954/Z/16/Z) and RS McDonald Fund) to M.T. (Wellcome Trust Multi-User Equipment grant (212947/Z/18/Z)) and A.G.S. (Epilepsy Research UK (F1603)). For the purpose of open access, the author has applied a CC-BY public copyright license to any author accepted manuscript version arising from this submission. We thank Stephen Mitchell for EM sample processing.

## Author contributions

Conceptualisation, K.B., M.A.C.; Methodology, K.B., K.L.D., M.P., M.S., E.B., A.G., E.C.D., M.T., A.G.S.; Formal Analysis, K.B., K.L.D., A.G., E.C.D., M.P., M.S., A.G.S.; Investigation, K.B., K.L.D., A.G.S., M.A.C.; Resources, M.A.C., M.T., A.G.S.; Writing—Original Draft, K.B., M.A.C.; Writing—Review & Editing, all authors; Funding Acquisition, M.A.C., M.T., A.G.S.

## Competing interests

The authors declare no competing interests.
