## [Peer Review File · Nature Communications]

REVIEWER COMMENTS

Reviewer #1 (Remarks to the Author):

This manuscript describes a new mouse model for DNM1 DEE and a possible new treatment for this ultrarare but devastating and intractable disease. The most important and exciting message is the suggestion of an existing stroke drug (BMSxxx) that might be effective in DNM1 DEE, or for other genetic NDD's where improving SV recycling may help. For reasons explained below I am not completely convinced that R237W is the best model from which to predict generalized success (even for other DNM1 DEE variants). However, I recognize that successful therapy for an intractable disease from a repurposed drug, even if partial, even if for only a subset of patients, would be most welcome.

The strongpoints are the assessment of the biochemical and synaptic vesicle defects, and the drug rescue of this and other apparent phenotypes in neurons from the mice or in vivo, and the general thoroughness and breadth of this study. A non-minor weak point is the relatively mild phenotype of these mice. The authors correctly recognize this is not unusual for mouse genetic NDD models, but it does question the effectiveness of this potential treatment for affected R237W children who have more severe impairments than the mice appear to (and maybe for other DNM1 variants that may be more severe or act by other mechanisms – e.g. impaired ring assembly). Another weakness is the likely indirect drug mechanism, through a combination of channels rather than on SV's directly - which the authors concede fairly enough, but again calls into question the precision of the therapy on the molecular defect (or relevance to other DNM1 molecular mechanisms let alone other NDDs). Still, at least some of the measured outcomes (e.g. EPSC amplitude) appear to rescue to wildtype levels and that is convincing for what it is.

While the results are formally novel, the authors should recognize relevant prior work done in the field. Papers describing Dnm1 fitful mice are the most relevant (the genetic DNM1 DEE field is tiny...and none were referenced or discussed – e.g. Boumil et al. 2010, Asinof et al. 2015, 2016; Dhindsa et al. 2015). Fitful (A408T) is a dominant-negative mutation in a mutually exclusive DNM1 alternate exon that encodes part of the dynamin-1 molecular assembly domain. This was the first de facto evidence that genetic defects in DNM1 are likely to be involved in epilepsy, proven true when patients with dominant-negative de novo variants in this and the GTPase domain first described several years later. Although this particular variant has not been seen yet in human DNM1, in 2022 Parthasarathy et al. described new DNM1 DEE variants also in the same alternate exon (absent from the 'other' DNM1 isoform), with disease features as severe as any prior described pathogenic variant and estimated to account for around 30% of all DNM1 cases (missed previously because this exon was not present in arrays used in clinical genetics). While the Parthasarathy paper should also be referenced for completeness, the point here is that it brings the Dnm1 fitful mouse and some results from the earlier papers clearly into the picture.

Compared to prior work, the current paper certainly has more assays, examines SV recycling directly, is more thorough and would be the first published (to my knowledge) animal model of a human DNM1 DEE variant. Whether it is because R237W is relatively mild cannot be known (but see the point further below). But, between Boumil et al. 2010 and Dhindsa et al. 2015 (including cell biology of 3 human variants, and EM showing SVs of fitful brain), there was a pretty clear smoking gun for a significant role of SV recycling defect in epilepsy (depleted vesicle pool upon repeated stimulation, impact on IPSC, defect in endocytosis, abnormal synapses by EM). Together these are all but a formal demonstration of “dysfunctional SV endocytosis in epilepsy.” Also, there is probably more from the earlier papers the authors could discuss, e.g. the relative impact on inhibition and IPSC. I don’t see that it detracts the authors’ very nice work to pull back on their ‘first’ claims and to find some relevant ways to compare to the prior work. They will still look good.

More importantly, certain comparisons will benefit the field’s understanding of the prospects of BMSxxx and what one can get out of and predict from different preclinical models. Although the authors couched the observations carefully, R237W mice appear to have electroclinical features and behaviors that could barely be termed seizures. In fitful while the severe DEE-like features with lethal early seizures were observed only in homozygotes, unlike R237W, fitful heterozygotes have actual generalized tonic-clonic seizures almost fully penetrant over a lifetime, including spontaneous (observed by EEG without handling). One wonders whether R237W is molecularly milder than the other variants (indeed it may be the most common variant for a reason), and possibly a relatively easier first test. Engineering workable mouse models is not easy. But the DNM1 cDNA is small enough for AAV or other manipulable viruses, to transduce wildtype primary neurons in culture. One could express multiple variants from either pathogenic domain and test the relative effectiveness of BMSxxx on mature cultures from different variants – it would be quite convincing if the drug worked the same or similar on different variants.

Reviewer #2 (Remarks to the Author):

In this study, the authors provide compelling evidence that the DNM1 heterozygous R237W mutation impairs vesicle endocytosis and synaptic transmission, and leads to epileptic encephalopathy. Both the impairments in SV endocytosis and the myoclonic jumping phenotype were effectively rescued by a clinical used drug BMS-204352. By using an elegant combination of endocytosis confocal imaging, electrophysiological recordings, EM imaging, and behaviors approaches, this work provides insight into the pathological role of patient derived point mutation of DNM1 R237W in synaptic transmission, and defines a novel mechanism mediating the pathogenesis of epileptic encephalopathy.

Overall, the findings are interesting and novel, the experimental design is rational and the conclusions are well supported by the data. However, the comments outlined below should be addressed by the authors prior to publication.

Major comments:

1. The time constant of sypHy endocytic decay (τ) should be quantified for a more precise evaluation of endocytic rate.
2. In contrast to the reduced RRP size and rate of vesicle pool replenishment in Dnm1+/R237W mice as revealed in Fig4h-j, the authors observed the increased excitatory transmission in Dnm1+/R237W synapses. Especially, they also reported the unchanged SV exocytosis with the sypHy assay. The authors should explain why the evoked EPSC and the release probability are increased with new experiments or a detailed discussion.
3. Since hyperactivity of hippocampal neurons has been co-correlated with epilepsy, in vivo or slice recordings of APs in Dnm1+/R237W mice should be included.
4. The authors provided jumping behaviors coupled with LFP recordings to verify the epilepsy phenotype of Dnm1+/R237W mice. However, patients with epilepsy often show profound intellectual disability, and the authors indeed observed the impaired synaptic endocytosis of hippocampal neurons. It would be useful to include the recordings of the LTP and learning & memory behaviors of Dnm1+/R237W mice.

Minor comments

1. Statistics of LFP recording traces, e.g. the frequency and duration of the burst, or energy spectrum analysis, should be included in Fig. 5.
2. The authors should provide typical electrophysiological recording traces for Figs. 4 and 7.
3. The scale bar in Fig. 3a is missing.
4. There seems to be a typo error under the x-axle in Fig. 3k.

Reviewer #3 (Remarks to the Author):

In the manuscript titled "Reversal of cell, circuit and seizure phenotypes in a mouse model of DNM1 epileptic encephalopathy", Bonnycastle, Cousin and colleagues investigate a common point mutation in dynamin-1, R237W, that is causative of an intractable form of epileptic encephalopathy. The authors show that R237W mutation impairs the GTPase activity of dynamin-1 and leads to reduced synaptic

vesicle endocytosis during high-frequency, prolonged activity. R237W mutation also has a small effect on neuron excitability, decreases spontaneous glutamate release and increases evoked probability of release in hippocampal CA3/CA1 synapses. The authors validate their animal model by showing that heterozygous mice carrying the R237W mutation have myoclonic jumping and increased spiking activity in different brain regions. They finally show that BMS-204352, a potassium (BK and KCNQ) channel “opener” (previously shown to revert abnormalities in fragile X syndrome and stroke), corrects all the phenotypes/dysfunctions associated with R237W mutation in dynamin-1.

The manuscript is well written, figures are easy to understand and data analysis is performed in a rigorous manner. The strength of this work is that it links the cellular and molecular changes in dynamin-1 to the physiological, circuit and mouse behavior abnormalities. The authors used a wide variety of techniques and model systems to validate and support their conclusions. The results obtained with BMS-204352 treatment are impressive and very promising, so I think that this research will be of interest for a broad audience and it could have long-term impact on epilepsy understanding and therapy development.

The mechanism of action of BMS-204352 remains a mystery, however, and it is not clear if the beneficial effects are via other targets, besides synaptic vesicle recycling. In my humble opinion, there are some missing links in the manuscript, even though it is clear that dynamin-1 R237W mutation leads to synaptic vesicle endocytosis deficits, it is not clear if this recycling effect is sufficient to cause the disease or if there are other underlying causes. For example, dynamin-1 endocytosis is crucial for BDNF-TrkB signaling at synapses, and it may also regulate the trafficking of membrane receptors and ion channels (even though the protein levels might not be changed, their subcellular distribution could be altered). I am aware that proving that synaptic vesicle endocytosis underlies the disease phenotype, by accelerating it for example, is technically challenging, and this manuscript already has a substantial amount of work. So I would only ask the authors to include these alternative options in the Discussion section.

It surprised me that only excitatory (AMPA-mediated) neurotransmission is tested in the manuscript, what about inhibitory neurotransmission, synapses and neurons? Reduced inhibitory tone could also contribute to the epilepsy, so it would be very informative to measure mIPSC and IPSC in the R237W heterozygous mice (as done in Figure 4).

Very minor comment: it would strengthen the interpretation if the experiments shown in Figure 3a are quantified.

Reviewer #4 (Remarks to the Author):

The Key Results of the research is the novel Dnm1 R237W mouse model and a novel potential therapeutic, already known to be safe in humans. The therapeutic shows promise in the Dnm1 R237W model; this is exciting and significant for the DNM1 EE field in general and certainly to the patients.

Some Major comments:

1. The data is valid, robust and the conclusions drawn from the data are all sound. The major flaw would be the lack of a seizure phenotype in the new model. It is a good model for the DNM1 field, but lacks the true recapitulation of an epileptic encephalopathy. There is another DNM1 model available that does have the hallmarks of the epileptic encephalopathy, including a well-documented seizure phenotype (PLoS Genet. 2010 Aug 5;6(8):e1001046.

doi: 10.1371/journal.pgen.1001046.)

Specific concerns:

Line 146: mice were fertile and were born in Mendelian proportions. What were the number of homozygotes that were born and lifespan?

Line 205: Misshapen SVs and endosomal-like compartments (Fig 3a). The number of SV's and the diameters should be measured and quantified statistically

Line 350: Bona fide seizure activity: myoclonic jumping is defined as "hind limb jumping in corner of cage with muzzle upward" per reference 30 Meidenbauer et al, and is a stereotypic and repetitive behavior. The jumping in the R237W mice may not indicate seizure like activity, but could be interpreted as a repetitive behavior as seen in models of autism. Interestingly, the DNM1 EE does overlap with some autistic traits. I would recommend to tone down the claim that the R237W mouse has seizure activity without further testing.

Lines 505: This statement is questionable. Epilepsy has not been demonstrated in this model

Lines 594-600. Please add the guide sequence used to create the model, and any off-target site predictions. Please add the number of backcrosses the modified were made to C57BL/6J mice. How many founder lines were generated? Were they maintained independently if more than one?

Minor Comments:

Lined 358: typo on small molecule #?

Significance

- 1.The Conclusions for the field and related fields is highly significant due to: Novel therapeutic
- 2.Other published findings support the significance of this research, but perhaps detract from the novelty. Specifically, the Dhindsa et al, Neurol Genet . 2015 Apr 17;1(1):e4.

doi: 10.1212/01.NXG.0000464295.65736.da. eCollection 2015 Jun. reference should be cited as a previous demonstration of DNM1 EE causing mutations that impair synaptic vesicle recycling.

Data and Methodology

Approach: valid

Data: good to high

Presentation quality: high

Enough detail in the M&M for work to be reproduced

Reviewer #1

This manuscript describes a new mouse model for DNM1 DEE and a possible new treatment for this ultrarare but devastating and intractable disease. The most important and exciting message is the suggestion of an existing stroke drug (BMSxxx) that might be effective in DNM1 DEE, or for other genetic NDD's where improving SV recycling may help. For reasons explained below I am not completely convinced that R237W is the best model from which to predict generalized success (even for other DNM1 DEE variants). However, I recognize that successful therapy for an intractable disease from a repurposed drug, even if partial, even if for only a subset of patients, would be most welcome.

The strongpoints are the assessment of the biochemical and synaptic vesicle defects, and the drug rescue of this and other apparent phenotypes in neurons from the mice or in vivo, and the general thoroughness and breadth of this study. A non-minor weak point is the relatively mild phenotype of these mice. The authors correctly recognize this is not unusual for mouse genetic NDD models, but it does question the effectiveness of this potential treatment for affected R237W children who have more severe impairments than the mice appear to (and maybe for other DNM1 variants that may be more severe or act by other mechanisms – e.g. impaired ring assembly).

We thank the reviewer for their helpful comments on our manuscript and we have discussed their concerns below.

Another weakness is the likely indirect drug mechanism, through a combination of channels rather than on SV's directly - which the authors concede fairly enough, but again calls into question the precision of the therapy on the molecular defect (or relevance to other DNM1 molecular mechanisms let alone other NDDs). Still, at least some of the measured outcomes (e.g. EPSC amplitude) appear to rescue to wildtype levels and that is convincing for what it is.

We would stress at this point that BMS-204352 corrects all phenotypes that we have examined to date. These include correction of: 1) dominant-negative effects of Dyn1_{R237W}-mCER on SV endocytosis in wild-type neurons (Figure 6a-c), 2) slowing of SV endocytosis in Dnm1^{+/R237W} neurons (Figure 6d-f), 3) evoked EPSC amplitude across a range of stimulus intensities (Figure 7d), 4) short-term plasticity of excitatory neurotransmission (Figure 7f) and, 5) the number of myoclonic jumps and bursts (Figure 8). Importantly, the neurotransmission and behavioural phenotypes in wild-type mice were unaffected by the drug (Figure 7b, 8). Therefore, while this drug may act via a combination of ion channels, the acceleration of SV endocytosis the key outcome. This therefore provides the precision noted by the reviewer in terms of potential therapy for a number of neurodevelopmental disorders that have defective SV endocytosis at their core.

While the results are formally novel, the authors should recognize relevant prior work done in the field. Papers describing Dnm1 fitful mice are the most relevant (the genetic DNM1 DEE field is tiny...and none were referenced or discussed – e.g. Boumil et al. 2010, Asinof et al. 2015, 2016; Dhindsa et al. 2015). Fitful (A408T) is a dominant-negative mutation in a mutually exclusive DNM1 alternate exon that encodes part of the dynamin-1 molecular assembly domain. This was the first de facto evidence that genetic defects in DNM1 are likely to be involved in epilepsy, proven true when patients with dominant-negative de novo variants in this and the GTPase domain first described several years later.

We were aware of the previous excellent work performed in the fitful mouse outlined above. The reason that we did not provide much emphasis to this model is that it carries a mutation that has never been identified in humans (as the reviewer acknowledges) and therefore the direct relevance to human disease is limited. It is acknowledged that mice homozygous for these mutations do

display spontaneous seizures, however the relevance of this model to the human condition is unclear, when one considers that all pathogenic mutations identified to date are heterozygous.

Regardless of these points, we have now discussed our results in the context of previous work performed in the fitful mouse (pages 11-12, lines 342-356, 360-362). We have not referred to the work involving mice that are either 1) heterozygous for the fitful mutation that have the remaining wild-type allele removed in specific cell types (the fitful floxed mouse - (Asinof et al., 2015) or 2) have a selective deletion of either Dnm1a or Dnm1b isoforms (Asinof et al., 2016). While this work is of high interest in terms of potential epileptogenic mechanisms, we felt it was too far removed from the human condition to be directly relevant to the work presented in this manuscript.

Although this particular variant has not been seen yet in human DNM1, in 2022 Parthasarathy et al. described new DNM1 DEE variants also in the same alternate exon (absent from the 'other' DNM1 isoform), with disease features as severe as any prior described pathogenic variant and estimated to account for around 30% of all DNM1 cases (missed previously because this exon was not present in arrays used in clinical genetics). While the Parthasarathy paper should also be referenced for completeness, the point here is that it brings the Dnm1 fitful mouse and some results from the earlier papers clearly into the picture.

The paper by Parthasarathy (2022) was referenced in the original manuscript as reference number 3. We have also now confirmed that we used the same middle domain splice variant in our overexpression studies that was demonstrated to be the predominant isoform in mammalian brain in this study (page 4, line 74-75).

Compared to prior work, the current paper certainly has more assays, examines SV recycling directly, is more thorough and would be the first published (to my knowledge) animal model of a human DNM1 DEE variant. Whether it is because R237W is relatively mild cannot be known (but see the point further below). But, between Boumil et al. 2010 and Dhindsa et al. 2015 (including cell biology of 3 human variants, and EM showing SVs of fitful brain), there was a pretty clear smoking gun for a significant role of SV recycling defect in epilepsy (depleted vesicle pool upon repeated stimulation, impact on IPSC, defect in endocytosis, abnormal synapses by EM). Together these are all but a formal demonstration of "dysfunctional SV endocytosis in epilepsy." Also, there is probably more from the earlier papers the authors could discuss, e.g. the relative impact on inhibition and IPSC. I don't see that it detracts the authors' very nice work to pull back on their 'first' claims and to find some relevant ways to compare to the prior work. They will still look good.

We apologise if we have given the impression of primacy in terms of the established relationship between dynamin-1 dysfunction (and by extension SV endocytosis) and epilepsy. As the reviewer correctly states, there have been a few prior studies examining either the effect of pathogenic DNM1 variants on endocytosis in heterologous expression systems or examining phenotypes in homozygous fitful mice.

In terms of discussing previous studies of inhibitory neurotransmission in homozygous fitful mice, we have now related our new experiments examining mIPSCs, evoked IPSCs and Pr (NEW Figure 4g-i, 4l, 4m, 4p, 4q) to both 1) the homozygous Dnm1 fitful and 2) the Dnm1 knockout mouse. This reveals that the effect of the R237W mutation has no effect on mIPSCs and evoked IPSCs (which is identical to acute slices from the homozygous fitful mouse (Boumil et al., 2010), but in contrast to primary neuronal cultures from the homozygous Dnm1 knockout (Ferguson et al., 2007). It should be noted that primary neurons from homozygous fitful mice display increased mEPSC/mIPSC

amplitudes (proposed to be due to larger SVs), and decreased evoked inhibitory, but not excitatory neurotransmission (McCabe et al., 2021).

In contrast to excitatory neurotransmission (which has a higher Pr) inhibitory neurotransmission has reduced Pr in Dnm1^{+/R237W} mice. This parameter has not been investigated to our knowledge in either Dnm1 fitful or Dnm1 knockout mice. The next phase of our research programme will be to perform rundown experiments to determine whether there is depression of inhibitory neurotransmission in a similar manner to that observed in the homozygous fitful mouse (Boumil et al., 2010).

More importantly, certain comparisons will benefit the field's understanding of the prospects of BMSxxx and what one can get out of and predict from different preclinical models. Although the authors couched the observations carefully, R237W mice appear to have electroclinical features and behaviors that could barely be termed seizures. In fitful while the severe DEE-like features with lethal early seizures were observed only in homozygotes, unlike R237W, fitful heterozygotes have actual generalized tonic-clonic seizures almost fully penetrant over a lifetime, including spontaneous (observed by EEG without handling). One wonders whether R237W is molecularly milder than the other variants (indeed it may be the most common variant for a reason), and possibly a relatively easier first test. Engineering workable mouse models is not easy. But the DNM1 cDNA is small enough for AAV or other manipulable viruses, to transduce wildtype primary neurons in culture. One could express multiple variants from either pathogenic domain and test the relative effectiveness of BMSxxx on mature cultures from different variants – it would be quite convincing if the drug worked the same or similar on different variants.

The reviewer proposes an interesting point with respect to the “mild” phenotype observed in Dnm1^{+/R237W} mouse, with the suggestion that it originates at the molecular level. We would be surprised if the R237W mutation was “milder” in molecular terms to other pathogenic variants, since as stated in the manuscript, this residue is critical in the co-ordination of GTP binding and stabilisation of this transition state during GTP hydrolysis (Chappie et al., 2010). To test this hypothesis, we expressed a mutation in a different domain of dynamin-1 that is purported to be pathogenic, the fitful mutation (A408T), in primary cultures of wild-type hippocampal neurons (new Extended Data Figure 1). Overexpression of this mutant has no significant effect on SV endocytosis using the same assays where pronounced defects had been observed with either the K44A or R237W mutants (Figure 1). Therefore the R237W mutant has a much more pronounced cellular phenotype than the fitful mutation defect in terms of dominant-negative effects on SV endocytosis. The absence of effect obviously precluded our ability to assess the effect of BMS-204352 on restoring function.

We were initially surprised by this result, however it must be considered that altered nerve terminal morphology or inhibitory neurotransmission has only ever been observed in homozygous, and not heterozygous, fitful mice (Boumil et al., 2010). In addition, it must be noted that very few studies examine the effect of dynamin-1 mutants in their natural context (in neurons). For example, almost all studies overexpress dynamin-1 mutants in heterologous cell lines where the dominant isoform is dynamin-2. Dynamin-1 mutants have much more pronounced effects in these non-neuronal cells when compared to central neurons in our study. This is exemplified by the relatively mild effect of the K44A mutant on SV endocytosis (which ablates receptor-mediated endocytosis in non-neuronal cells) and the absence of effect of the A408T mutant (which inhibits receptor-mediated endocytosis when overexpressed in COS-7 cells, (Dhindsa et al., 2015). We have

now included a discussion on the importance of examining dynamin-1 mutants in their natural context in the results section (page 12, line 350-356).

Reviewer #2

In this study, the authors provide compelling evidence that the DNM1 heterozygous R237W mutation impairs vesicle endocytosis and synaptic transmission, and leads to epileptic encephalopathy. Both the impairments in SV endocytosis and the myoclonic jumping phenotype were effectively rescued by a clinical used drug BMS-204352. By using an elegant combination of endocytosis confocal imaging, electrophysiological recordings, EM imaging, and behaviors approaches, this work provides insight into the pathological role of patient derived point mutation of DNM1 R237W in synaptic transmission, and defines a novel mechanism mediating the pathogenesis of epileptic encephalopathy.

Overall, the findings are interesting and novel, the experimental design is rational and the conclusions are well supported by the data. However, the comments outlined below should be addressed by the authors prior to publication.

We thank the reviewer for their positive comments and we have addressed their concerns below.

Major comments:

1. The time constant of sypHy endocytic decay (τ) should be quantified for a more precise evaluation of endocytic rate.

We attempted to produce this value for the reviewer, however in most cases there were instances where τ values could not be calculated accurately (due to a number of pHluorin traces in each set of experiments not conforming to first-order kinetics). It was for this reason that we used the distance to baseline calculation in the original manuscript, since we did not want to exclude outliers. We have added a statement in the methods section explaining this approach (page 17, line 515-516).

2. In contrast to the reduced RRP size and rate of vesicle pool replenishment in Dnm1^{+/R237W} mice as revealed in Fig4h-j, the authors observed the increased excitatory transmission in Dnm1^{+/R237W} synapses. Especially, they also reported the unchanged SV exocytosis with the sypHy assay. The authors should explain why the evoked EPSC and the release probability are increased with new experiments or a detailed discussion.

The reviewer raises an interesting point regarding apparent dichotomy between increases in both Pr and evoked EPSCs compared to decreased RRP in Dnm1^{+/R237W} slices. From first principles, the decrease in RRP size and its replenishment during action potential trains would be a consequence of perturbed SV endocytosis. Because of this, we were somewhat surprised by the increase in both Pr and evoked EPSCs. This suggests that the presence of the mutant Dnm1 R237W allele is impacting neurotransmitter release at the level of the fusion of a single SV, since EPSCs were evoked in both cases by a single 50 μ s pulse. This could be due to direct effects of the R237W allele on the SV fusion machinery (there is no published evidence of a role for dynamin-1 in this respect), modulation of ion channel function, or an indirect effect on other presynaptic signalling mechanisms. We have now discussed the latter two points in the discussion section, in terms of either AP broadening or altered trafficking of modulatory receptors at the presynapse (pages 13-14, lines 396-400, 406-413).

The reviewer is correct in noting the lack of a significant effect on the AP-evoked sypHy peak in Dnm1^{+/R237W} neurons (Figures 3 and 6). The difference between these results and those from the electrophysiological approaches are almost certainly due to the different outputs of these assays.

Both are highly informative when used appropriately, however both have limitations. For example, pHluorin reporters do not have the temporal resolution of electrophysiology, which can detect SV fusion events in response to single APs. In contrast, pHluorin imaging reports SV fusion in response to AP trains, without contamination from potential postsynaptic effects. Due to the time resolution of this assay being at least two orders of magnitude slower than electrophysiological monitoring, it is highly likely that the increased release observed in the EPSC increase is lost in the noise of the assay. This is especially pertinent when one considers that potential increases in SV fusogenicity might be offset by EPSC rundown during AP trains (Figure 4s).

3. Since hyperactivity of hippocampal neurons has been co-correlated with epilepsy, in vivo or slice recordings of APs in Dnm1+/R237W mice should be included.

This is an excellent suggestion. We have already performed an extensive analysis AP properties of Dnm1^{+/R237W} slices in comparison to Dnm1^{+/+}, however it was placed in the intrinsic properties section of the manuscript in Extended Data Table S2. We have now provided this data in the main manuscript (Figure 4a-c). We found that the excitability of Dnm1^{+/R237W} CA1 neurons were not significantly different in terms of either their spike frequency or threshold to fire APs, when compared to Dnm1^{+/+} controls. Therefore the alterations in excitatory neurotransmission observed in Dnm1^{+/R237W} mice is likely a consequence of presynaptic mechanisms, rather than intrinsic hyperactivity of individual neurons. We have now referred to this point in the main manuscript (page 7, line 186-189).

4. The authors provided jumping behaviors coupled with LFP recordings to verify the epilepsy phenotype of Dnm1+/R237W mice. However, patients with epilepsy often show profound intellectual disability, and the authors indeed observed the impaired synaptic endocytosis of hippocampal neurons. It would be useful to include the recordings of the LTP and learning & memory behaviors of Dnm1+/R237W mice.

The reviewer is correct regarding the co-morbidity between epilepsy and intellectual disability / autism. Indeed, many patients with DNM1 mutations display profound developmental delay and intellectual disability. We agree that investigating whether Dnm1^{+/R237W} mice display deficiencies in both learning and memory is an important question to address, however in the context of this manuscript we feel that the additional work required to thoroughly investigate this is outside its current scope.

We also suggest that this is the case for the examination of LTP, since in this particular manuscript we are focusing on exclusively presynaptic mechanisms, whereas the induction and maintenance of CA3-CA1 LTP is exclusively postsynaptic in nature. We agree that there may be potential defects in this process (although the amplitude of mEPSCs appears normal, suggesting that there is no obvious defect in the number of ionotropic glutamate receptors at the postsynaptic membrane), however we feel that once again this is outside the current scope of this manuscript.

Minor comments

1. Statistics of LFP recording traces, e.g. the frequency and duration of the burst, or energy spectrum analysis, should be included in Fig. 5.

This information has now been provided (NEW Figure 5c-e). The analysis reveals that there is an increase in the energy spectrum in Dnm1^{+/R237W} mice during myoclonic jumping across all bands. This is not due to increased neuromuscular activity since, 1) we compared power to Dnm1^{+/+} controls during their jumping activity, and 2) there was a significant increase in the lower frequency bands of delta and theta, which also provide confidence that the increased power is

not due to due to muscular activity during jumping. Furthermore, the duration of electrophysiological activity is also two fold greater during myoclonic jumping in $Dnm1^{+/R237W}$ mice compared to $Dnm1^{+/+}$ controls (NEW Figure 5e). It should be noted that the data from $Dnm1^{+/+}$ mice are $n=2$ animals (although the power spectrum of multiple jumping events were sampled for every mouse), since few of these mice displayed this jumping behaviour.

2. The authors should provide typical electrophysiological recording traces for Figs. 4 and 7.
This has now been provided (NEW Figure 4 and 7).

3. The scale bar in Fig. 3a is missing.
This has been corrected.

4. There seems to be a typo error under the x-axle in Fig. 3k.
This has been corrected.

Reviewer #3

In the manuscript titled “Reversal of cell, circuit and seizure phenotypes in a mouse model of DNM1 epileptic encephalopathy”, Bonnycastle, Cousin and colleagues investigate a common point mutation in dynamin-1, R237W, that is causative of an intractable form of epileptic encephalopathy. The authors show that R237W mutation impairs the GTPase activity of dynamin-1 and leads to reduced synaptic vesicle endocytosis during high-frequency, prolonged activity. R237W mutation also has a small effect on neuron excitability, decreases spontaneous glutamate release and increases evoked probability of release in hippocampal CA3/CA1 synapses. The authors validate their animal model by showing that heterozygous mice carrying the R237W mutation have myoclonic jumping and increased spiking activity in different brain regions. They finally show that BMS-204352, a potassium (BK and KCNQ) channel “opener” (previously shown to revert abnormalities in fragile X syndrome and stroke), corrects all the phenotypes/dysfunctions associated with R237W mutation in dynamin-1.

The manuscript is well written, figures are easy to understand and data analysis is performed in a rigorous manner. The strength of this work is that it links the cellular and molecular changes in dynamin-1 to the physiological, circuit and mouse behavior abnormalities. The authors used a wide variety of techniques and model systems to validate and support their conclusions. The results obtained with BMS-204352 treatment are impressive and very promising, so I think that this research will be of interest for a broad audience and it could have long-term impact on epilepsy understanding and therapy development.

We thank the reviewer for their generous comments on the work.

The mechanism of action of BMS-204352 remains a mystery, however, and it is not clear if the beneficial effects are via other targets, besides synaptic vesicle recycling. In my humble opinion, there some missing links in the manuscript, even though it is clear that dynamin-1 R237W mutation leads to synaptic vesicle endocytosis deficits, it is not clear if this recycling effect is sufficient to cause the disease or if there are other underlying causes. For example, dynamin-1 endocytosis is crucial for BDNF-TrkB signaling at synapses, and it may also regulate the trafficking of membrane receptors and ion channels (even though the protein levels might not be changed, their subcellular distribution could be altered). I am aware that proving that synaptic vesicle endocytosis underlies the disease phenotype, by accelerating it for example, is technically challenging, and this manuscript already has a substantial amount of work. So I would only ask the authors to include these alternative options in

the Discussion section.

We thank the reviewer for their understanding regarding the precise mechanism of action of BMS-204352 and obtaining direct causal evidence of defects in SV endocytosis and epilepsy. We have now added in a paragraph in the discussion section which covers other potential causes of the observed in vivo phenotypes in the $Dnm1^{+/R237W}$ mouse. These include, altered dynamin-dependent trafficking of ion channels, postsynaptic receptors and modulatory presynaptic receptors (pages 13-14, lines 389-413).

It surprised me that only excitatory (AMPA-mediated) neurotransmission is tested in the manuscript, what about inhibitory neurotransmission, synapses and neurons? Reduced inhibitory tone could also contribute to the epilepsy, so it would be very informative to measure mIPSC and IPSC in the R237W heterozygous mice (as done in Figure 4).

The reviewer makes an excellent point, particularly in light of the disproportionate effect on inhibitory neurotransmission observed in $Dnm1$ knockout mice (Ferguson et al., 2007). We have now provided new data that measures mIPSC frequency / amplitude, the paired-pulse ratio (PPR) and the amplitude of evoked IPSCs (NEW Figure 4g-i, 4l, 4m, 4p, 4q). These results are intriguing, since we see no change in either mIPSC frequency or in evoked IPSCs in slices from $Dnm1^{+/R237W}$ mice when compared to wild-type controls. This is in contrast to the $Dnm1$ knockout mouse (Ferguson et al., 2007), but consistent with the homozygous fitful mouse (which only displays defects in inhibitory neurotransmission during a train of action potential stimulation (Boumil et al., 2010), please see extended response to referee 1 above). Intriguingly, mIPSC amplitude is larger in $Dnm1^{+/R237W}$ mice, suggesting either increased surface expression of GABA_A receptors (discussed in the context of the point above) or larger SVs due to $Dnm1$ dysfunction (as proposed for the $Dnm1$ knockout mouse, (Ferguson et al., 2007, Hayashi et al., 2008). The increased size of SVs in $Dnm1^{+/R237W}$ nerve terminals (new Figure 3d), suggests the latter may be the case.

We did observe a change in the PPR at inhibitory $Dnm1^{+/R237W}$ synapses, and it was the converse of the excitatory PPR (NEW Figure 4p,4q). Therefore $Dnm1^{+/R237W}$ circuits have excitatory synapses with increased Pr, and inhibitory synapses with decreased Pr, providing a potential mechanistic explanation for the hyperexcitable phenotypes observed in $Dnm1^{+/R237W}$ mice. This has now been referred to in the discussion (page 12, lines 373-376).

Very minor comment: it would strengthen the interpretation if the experiments shown in Figure 3a are quantified.

We have now performed this analysis (NEW Figure 3b-d, Extended Data 3a). As suggested by the images shown in the original version of the manuscript, there was a large increase in the number of presynaptic endosomes in the nerve terminals of $Dnm1^{+/R237W}$ mice (an approximate increase of 250%, Figure 3b). Additionally, there was an increase in the overall size of the remaining SVs within these nerve terminals (Figure 3d). Both of these are indicative of dysfunctional endocytosis, which was confirmed by our subsequent analyses using both sypHy and HRP uptake assays (remainder of figure 3). Interestingly, there was a small, but significant, increase in the number of SVs in $Dnm1^{+/R237W}$ nerve terminals (Figure 3c). The scale of the increase was much smaller than the number of endosomes (17% vs 250% respectively), and future experiments will investigate this phenomenon further.

Reviewer #4

The Key Results of the research is the novel $Dnm1$ R237W mouse model and a novel potential therapeutic, already known to be safe in humans. The therapeutic shows promise in the $Dnm1$

R237W model; this is exciting and significant for the DNM1 EE field in general and certainly to the patients.

We thank the reviewer for their generous comments on the potential significance of the work.

Some Major comments:

1. The data is valid, robust and the conclusions drawn from the data are all sound. The major flaw would be the lack of a seizure phenotype in the new model. It is a good model for the DNM1 field, but lacks the true recapitulation of an epileptic encephalopathy. There is another DNM1 model available that does have the hallmarks of the epileptic encephalopathy, including a well-documented seizure phenotype (PLoS Genet. 2010 Aug 5;6(8):e1001046. doi: 10.1371/journal.pgen.1001046.)

We thank the reviewer for highlighting this and we have now discussed the findings from the fitful mouse model in the context of our current work in the discussion section (pages 11-12, lines 342-356, 360-362).

Specific concerns:

Line 146: mice were fertile and were born in Mendelian proportions. What were the number of homozygotes that were born and lifespan?

Our breeding strategy was to only breed heterozygotes with wild-type mice, to generate a 50:50 distribution of $Dnm1^{+/R237W}$ and $Dnm1^{+/+}$ mice. Therefore this is what we referred to in the manuscript with respect to Mendelian proportions. We have now clarified this in the text (page 15, line 457-458). We have never attempted to cross $Dnm1^{+/R237W}$ mice (to potentially generate $Dnm1^{R237W/R237W}$ mice), since the predicted severity would exceed our current Home Office licence guidelines. In a disease context, all pathogenic DNM1 mutations are heterozygous, therefore we felt that the homozygous condition did not have an obvious construct validity (in addition to the ethical licence issues discussed above).

Line 205: Misshapen SVs and endosomal-like compartments (Fig 3a). The number of SV's and the diameters should be measured and quantified statistically

We have now performed this analysis and it is now presented in Figure 3b-d (and Extended Data 3a). Please also see our response to an identical request from referee 3 (last point addressed).

Line 350: Bona fide seizure activity: myoclonic jumping is defined as "hind limb jumping in corner of cage with muzzle upward" per reference 30 Meidenbauer et al, and is a stereotypic and repetitive behavior. The jumping in the R237W mice may not indicate seizure like activity, but could be interpreted as a repetitive behavior as seen in models of autism. Interestingly, the DNM1 EE does overlap with some autistic traits. I would recommend to tone down the claim that the R237W mouse has seizure activity without further testing.

We acknowledge this point and have reduced the emphasis on this statement (page 11, line 336).

Lines 505: This statement is questionable. Epilepsy has not been demonstrated in this model

We have toned this statement down in the revised manuscript and in the revised abstract (page 2, line 35; page 11, line 328).

Lines 594-600. Please add the guide sequence used to create the model, and any off-target site predictions. Please add the number of backcrosses the modified were made to C57BL/6J mice. How many founder lines were generated? Were they maintained independently if more than one?

The requested information has now been added in the methods section (page 15, lines 452-458). As noted in the methods, two founder lines were generated, and one was taken forward to

establish the colony. The colony was backcrossed 3 times and has been routinely backcrossed 3 times each 5 generations to refresh it. Off-site target prediction were made by the company that generated the mouse, however these were not tested.

Minor Comments:

Lined 358: typo on small molecule #?

Corrected.

Reference List

- ASINOF, S., MAHAFFEY, C., BEYER, B., FRANKEL, W. N. & BOUMIL, R. 2016. Dynamin 1 isoform roles in a mouse model of severe childhood epileptic encephalopathy. *Neurobiol Dis*, 95, 1-11.
- ASINOF, S. K., SUKOFF RIZZO, S. J., BUCKLEY, A. R., BEYER, B. J., LETTS, V. A., FRANKEL, W. N. & BOUMIL, R. M. 2015. Independent Neuronal Origin of Seizures and Behavioral Comorbidities in an Animal Model of a Severe Childhood Genetic Epileptic Encephalopathy. *PLoS Genet*, 11, e1005347.
- BOUMIL, R. M., LETTS, V. A., ROBERTS, M. C., LENZ, C., MAHAFFEY, C. L., ZHANG, Z. W., MOSER, T. & FRANKEL, W. N. 2010. A missense mutation in a highly conserved alternate exon of dynamin-1 causes epilepsy in fitful mice. *PLoS Genet*, 6.
- CHAPPIE, J. S., ACHARYA, S., LEONARD, M., SCHMID, S. L. & DYDA, F. 2010. G domain dimerization controls dynamin's assembly-stimulated GTPase activity. *Nature*, 465, 435-40.
- DHINDSA, R. S., BRADRICK, S. S., YAO, X., HEINZEN, E. L., PETROVSKI, S., KRUEGER, B. J., JOHNSON, M. R., FRANKEL, W. N., PETROU, S., BOUMIL, R. M. & GOLDSTEIN, D. B. 2015. Epileptic encephalopathy-causing mutations in DNM1 impair synaptic vesicle endocytosis. *Neurol Genet*, 1, e4.
- FERGUSON, S. M., BRASNJO, G., HAYASHI, M., WÖLFEL, M., COLLESI, C., GIOVEDI, S., RAIMONDI, A., GONG, L. W., ARIEL, P., PARADISE, S., O'TOOLE, E., FLAVELL, R., CREMONA, O., MIESENBOCK, G., RYAN, T. A. & DE CAMILLI, P. 2007. A selective activity-dependent requirement for dynamin 1 in synaptic vesicle endocytosis. *Science*, 316, 570-4.
- HAYASHI, M., RAIMONDI, A., O'TOOLE, E., PARADISE, S., COLLESI, C., CREMONA, O., FERGUSON, S. M. & DE CAMILLI, P. 2008. Cell- and stimulus-dependent heterogeneity of synaptic vesicle endocytic recycling mechanisms revealed by studies of dynamin 1-null neurons. *Proc Natl Acad Sci U S A*, 105, 2175-80.
- MCCABE, M. P., SHORE, A. N., FRANKEL, W. N. & WESTON, M. C. 2021. Altered Fast Synaptic Transmission in a Mouse Model of DNM1-Associated Developmental Epileptic Encephalopathy. *eNeuro*, 8.

REVIEWERS' COMMENTS

Reviewer #1 (Remarks to the Author):

The authors have done a very nice job both replying to my comments (in rebuttal or in the manuscript) or extending their work or discussion based on some of my suggestions (e.g. relation to inhibition). I thought this was initially an exciting manuscript and set of results, and even moreso now and i especially appreciate the authors' forbearance and vigilance.

I am still not certain (for some of the reasons the authors provide) on the severity of the respective variants and thus prospects for impact of the drug. For example, while for a number of reasons A408T (mouse 'fitful' variant) is not of primacy here, the test of allele strength may not bear out in vivo in native context especially if a key effect is in interneurons during early postnatal development. Still the rebuttal comments and revisions further reflect the outstanding work that the authors are doing. I look forward to seeing this in press.

Reviewer #2 (Remarks to the Author):

The authors provided a thorough response to the reviewers' comments, providing the addition of several experiments, analyses, and discussion. The new experimental data and more information included in the revision have strengthened the manuscript enormously. I have no further comments.

Reviewer #3 (Remarks to the Author):

The authors have nicely and convincingly addressed all the issues raised by the reviewers and the manuscript is significantly improved. It is my opinion that the work is ready for publication and I have no further comments. Thanks to the authors for their careful replies and hard work.

Reviewer #4 (Remarks to the Author):

The authors have addressed all the comments I provided very nicely. I think that this is a very important piece of research that the Dnm1 community, especially the patients, will be excited to learn about and give hope to a therapeutic. I fully support and encourage the publication of the manuscript in Nature Communications.

We thank all of the reviewers for their insightful comments on the manuscript in the previous round of revisions. Their input has improved the work significantly. There are no further points to address from all four reviewers.

Reviewer #1 (Remarks to the Author):

The authors have done a very nice job both replying to my comments (in rebuttal or in the manuscript) or extending their work or discussion based on some of my suggestions (e.g. relation to inhibition). I thought this was initially an exciting manuscript and set of results, and even more so now and I especially appreciate the authors' forbearance and vigilance.

I am still not certain (for some of the reasons the authors provide) on the severity of the respective variants and thus prospects for impact of the drug. For example, while for a number of reasons A408T (mouse 'fitful' variant) is not of primacy here, the test of allele strength may not bear out in vivo in native context especially if a key effect is in interneurons during early postnatal development. Still the rebuttal comments and revisions further reflect the outstanding work that the authors are doing. I look forward to seeing this in press.

Reviewer #2 (Remarks to the Author):

The authors provided a thorough response to the reviewers' comments, providing the addition of several experiments, analyses, and discussion. The new experimental data and more information included in the revision have strengthened the manuscript enormously. I have no further comments.

Reviewer #3 (Remarks to the Author):

The authors have nicely and convincingly addressed all the issues raised by the reviewers and the manuscript is significantly improved. It is my opinion that the work is ready for publication and I have no further comments. Thanks to the authors for their careful replies and hard work.

Reviewer #4 (Remarks to the Author):

The authors have addressed all the comments I provided very nicely. I think that this is a very important piece of research that the Dnm1 community, especially the patients, will be excited to learn about and give hope to a therapeutic. I fully support and encourage the publication of the manuscript in Nature Communications.